

# Lipid remodeling in phytoplankton exposed to multi-environmental drivers in a mesocosm experiment

Sebastian I. Cantarero[1*], Edgart Flores[1], Harry Allbrook[1], Paulina Aguayo[2,3,4], Cristian A. Vargas[3,5], John E. Tamanaha[6], J. Bentley C. Scholz[6], Lennart T. Bach[7], Carolin R. Löscher[8,9], Ulf Riebesell[10], Balaji Rajagopalan[11], Nadia Dildar[1], Julio Sepúlveda[1,5]

[1]Deparment of Geological Sciences and Institute of Arctic and Alpine Research (INSTAAR), University of Colorado Boulder, Boulder, CO 80309, USA
[2]Departamento de Oceanografía, Universidad de Concepción, Casilla 160-C, Concepción, Chile
[3]Department of Aquatic System, Faculty of Environmental Sciences & Environmental Sciences Center EULA Chile, Universidad de Concepción, Concepción 4070386, Chile
[4]Institute of Natural Resources, Faculty of Veterinary Medicine and Agronomy, Universidad de Las Américas, Sede Concepcion, Chacabuco 539, Concepcion 3349001, Chile
[5]Millennium Institute of Oceanography (IMO), Universidad de Concepción, Concepción 4070386, Chile
[6]Laboratory for Interdisciplinary Statistical Analysis, Department of Applied Mathematics, University of Colorado, Boulder, Boulder CO, 80309, USA
[7]Institute for Marine and Antarctic Studies, University of Tasmania, Hobart, TAS 7004, Australia
[8]Nordcee, Department of Biology, University of Southern Denmark, Odense, Denmark
[9]Danish Institute for Advanced Study, University of Southern Denmark, Odense, Denmark
[10]Marine Biogeochemistry, GEOMAR Helmholtz Centre for Ocean Research Kiel, Düsternbrooker Weg 20, D-24105, Kiel, Germany
[11]Department of Civil, Environmental, and Architectural Engineering, University of Colorado-Boulder, Boulder, CO, USA

*Correspondence to*: Sebastian I. Cantarero (sebastian.cantarero@colorado.edu)

*Now at Vesta Public Benefit Corporation

**Abstract.** Lipid remodeling, the modification of cell membrane chemistry via structural rearrangements within the lipid pool of an organism, is a common physiological response amongst all domains of life to alleviate environmental stress and maintain cellular homeostasis. Whereas culture experiments and environmental studies of phytoplankton have demonstrated the plasticity of lipids in response to specific abiotic stressors, few analyses have explored the impacts of multi-environmental stressors at the community-level scale. Here, we study changes in the pool of intact polar lipids (IPLs) of a phytoplanktonic community exposed to multi-environmental stressors during a ~2-month long mesocosm experiment deployed in the eastern tropical South Pacific off the coast of Callao, Perú. We investigate lipid remodeling of IPLs in response to changing nutrient stoichiometries, temperature, pH, and light availability in surface and subsurface water-masses with contrasting redox potentials, using multiple linear regressions, classification and regression trees, and Random Forest analyses. Notable responses include the proportional increases of certain glycolipids (namely mono- and di-galactosyldiacylglyercols; MG and DG, respectively) associated with thermal stress as well as the degradation of these lipids under oxygen stress. Reduced $CO_2(aq)$ availability and high pH are associated with increased DG and sulfoquinovosyldiacylglycerol (SQ) concentrations. Higher production of MG in surface waters corresponds well with their stablished photoprotective and antioxidant mechanisms in thylakoid membranes. Certain phosphatidylglycerol (PG) moieties show strong linear trends with light availability and are known to be important components in electron transport processes of photosystems I and II. IPL remodeling suggests the





recycling of acyl chains for energy storage in the form of triacylglycerols (TAGs) in response to stressors; like N limitation,
variable pH, hypoxia, and photoinhibition. These physiological responses reallocate resources from structural or
extrachloroplastic membrane lipids (i.e., phospholipids and betaine lipids) under high-growth conditions, to thylakoid/plastid
membrane lipids (i.e., glycolipids and certain PGs) and TAGs under growth-limiting conditions. Investigation of this lipid
remodeling system is necessary to understand how membrane reorganization can affect the pools of cellular C, N, and S, and
how it may influence fluxes of biologically relevant elements to higher trophic levels and to the dissolved organic matter pool.

## 1 Introduction

The Eastern Tropical South Pacific (ETSP) is one of the most productive eastern boundary upwelling systems in the world
(Chavez and Messié, 2009) and harbors one of the largest oxygen deficient zones (ODZs) (Fuenzalida et al., 2009; Ulloa and
Pantoja, 2009; Thamdrup et al., 2012). Global warming has led to the expansion of ODZs over the recent decades and they are
expected to continue expanding due to the reduction in oxygen solubility with increasing temperature (Stramma et al., 2008;
Stramma et al., 2010; Gilly et al., 2013), as well as because of enhanced ocean stratification and reduced ventilation of the
ocean's interior (Keeling et al., 2010). The future behavior of the ETSP upwelling system in a warmer world remains uncertain;
increases in wind-induced upwelling intensity and duration (Gutiérrez et al., 2011; Bakun et al., 2010) may increase the supply
of nutrients to the surface in coastal regions, whereas enhanced thermal stratification may reduce nutrient supply in the open
ocean (Behrenfeld et al., 2006). Furthermore, upwelling regions are prone to highly variable pH (Capone and Hutchins, 2013),
and the global ocean will experience a decreasing average pH as more $CO_2$ accumulates in the atmosphere and is absorbed by
the ocean (Jiang et al., 2019). Accordingly, major shifts in marine planktonic community composition, turnover rates (Henson
et al., 2021), and adaptations (Irwin et al., 2015) are expected in future scenarios of ocean conditions, which are expected to
lead to cascading effects on ocean biogeochemistry and marine ecosystems (Hutchins and Fu, 2017).

Primary productivity in the ETSP is predominantly regulated by the wind-induced upwelling of nutrients, light availability,
and Fe limitation (Messié and Chavez, 2015). Thus, changes in the supply of inorganic N along the upwelling region of the
ETSP are likely to induce significant shifts in the phytoplankton community composition. Longer upwelling seasons along
nearshore environments could further stimulate productivity of fast-growing eukaryotic algae that currently dominate these
systems (e.g., diatoms; Messié et al., 2009). However, shorter upwelling seasons or weaker upwelling currents could favor
more survivalist or mixotrophic algae, in addition to N-fixing diazotrophs that thrive under widespread nitrogen limitation
(Dutkiewicz et al., 2012). The rate of primary productivity in the surface ocean not only affects the supply of sinking organic
matter and thus oxygen consumption via microbial respiration in the subsurface (Wyrtki, 1962), but also results in a shift of
redox potentials that drive substantial losses of bioavailable N under reducing conditions at intermediate depths (Lam and
Kuypers, 2009; Wright et al., 2012). Additionally, expected changes in ocean warming and stratification (Huertas et al., 2011;
Morán et al., 2010; Yvon-Durocher, 2015), lowered dissolved oxygen concentration (Wu et al., 2012; Catalanotti et al., 2013),
and decreased pH (Dutkiewicz et al., 2015, Bach et al., 2017), will disrupt phytoplanktonic assemblages differently based on



their individual tolerances and physiological plasticity. However, little is known of the physiological adaptations of phytoplankton on a community-level scale in response to multi environmental stressors.

Phytoplankton have been shown to activate several lipid-based physiological mechanisms in response to environmental stimuli (Li-Beisson et al., 2019; Sayanova et al., 2017; Kong et al., 2018). The remodeling of intact polar lipids (IPLs), the main constituents of cell and chloroplast membranes, provides numerous physiological adjustments to attenuate environmental stressors impacting phytoplankton (Zienkiewicz et al., 2016; Du and Benning 2016; Morales et al., 2021). These include nutrient limitation (Van Mooy et al., 2009; Meador et al., 2017; Abida et al., 2015; Urzica et al., 2013; Gordillo et al., 1998; Wang et al., 2016), homeoviscous regulation in response to changing temperature (Sato et al., 1979; Wada and Murata, 1990; Sinensky, 1974; Neidleman, 1987; Tatsusawa and Takizawa 1996) and pH (Tatsusawa and Takizawa 1996; Poerschmann et al., 2004; Guckert and Cooksey, 1990), or photosynthetic function under varying light availability (Sato et al., 2003; Gombos et al., 2002; Pineau et al., 2004; Simionato et al., 2013; Gašparović et al., 2013; Khotimchenko and Yakovleva, 2005). While IPL distributions in environmental studies are typically used as chemotaxonomic biomarkers that trace the presence and abundance of specific microbial groups (Sturt et al., 2004; Schubotz et al., 2009; Van Mooy and Fredricks, 2010), their distributions have been used to in conjunction with additional microbial or geochemical measurements to assess how microbial metabolisms contribute to the chemical environment (Van Mooy et al., 2009; Wakeham et al., 2012; Schubotz et al., 2018; Cantarero et al., 2020). Yet, few studies have explored how multiple environmental drivers impact IPL remodeling at the community level and in time series, s, and the associated adaptability of phytoplankton to environmental change.

Lab-based culture experiments have been a major step forward in understanding how lipid remodeling may impact a biogeochemical system (as summarized above). However, a significant challenge remains in contextualizing these findings at the community scale. Conversely, observational studies from direct measurements of natural systems are often logistically limited in temporal scale, and consequently, do not fully capture the dynamics and heterogeneity of biogeochemical conditions. Mesocosms are experimental apparatuses at the interface between controlled culture experiments and environmental observations that allow for the examination of natural systems and entire ecosystems under semi-controlled conditions to explore the impacts of a changing climate and ocean system (Riebesell et al., 2013). Here, we study changes in the composition, diversity, and abundance of phytoplanktonic IPLs in response to changes in the biological, physical, and chemical composition of a marine ecosystem subjected to semi-controlled conditions in a 2-month long mesocosm experiment off the coast of Perú. We investigate the potential for IPL remodeling amongst phytoplankton in response to multiple environmental stressors including nutrient availability, $O_2$ concentration, pH, temperature, and light availability to highlight adaptation strategies available to phytoplankton in response to a changing ocean system.





## 2 Methods

### 2.1 Mesocosm Deployment and Sampling

On February 22, 2017, eight "Kiel Off-Shore Mesocosms for Ocean Simulations" (KOSMOS; Riebesell et al., 2013) were deployed just north of Isla San Lorenzo, 6 km off the Peruvian coastline (12.0555º S, 77.2348º W; ~30 m water depth; Fig. 1; Bach et al., 2020). Each mesocosm consisted of a cylindrical polyurethane bag (2 m diameter, 18.7 m long, 54.4 ±1.3 m³ volume) suspended in an 8 m tall flotation frame. Nets (mesh size 3 mm) attached to both ends of the bags allowed water exchange for three days before the water mass inside each mesocosm was enclosed and isolated from the surrounding Pacific water by attaching sediment traps at the base (~19 m). The enclosing of the mesocosms marked the start (day 0) of the 50-day experiment. We sampled two integrated water depths of the mesocosms for suspended organic matter and biological and chemical characterization using 5-L integrating water samplers (IWSs; Hydro-Bios, Kiel) equipped with pressure sensors to collect water evenly within a desired depth range. These depth ranges were adjusted over time to ensure capture of the low oxygen subsurface water as the oxycline slightly shifted over the course of the experiment. From days 1-2 the depth ranges were 0–5 and 5–17 m, from days 3 to 28 they were 0–10 and 10–17 m, and finally 0–12.5 and 12.5– 17 m from days 29 to 50. Sampling of chl-a and physicochemical and biogeochemical conditions was performed on all 8 mesocosms as well as in the control Pacific water outside of the mesocosms (Bach et al., 2020).

As detailed in Bach et al. (2020), the experiment involved several manipulations including the addition of ODZ water to simulate upwelling conditions, salt to maintain water stratification, and the introduction of organisms. On days 5 and 10 of the experiment, two-100 m³ batches of local ODZ water was collected at a depth of 30 m and 70 m from stations 1 (12.028323° S, 77.223603° W; very low N:P ratio) and 3 (12.044333° S, 77.377583° W; low N:P ratio), respectively, using deep water collectors described by Taucher et al. (2017). Bach et al. (2020) provides a detailed explanation of the deep water addition. Briefly, water in M1, M4, M5, and M8 (~20 m³) was mixed with ODZ water from station 1, while water in M2, M3, M6, and M7 was mixed with ODZ water from station 3 (see Table 1 for nutrient stoichiometries). To maintain stratification and preserve the low O₂ subsurface layer, a NaCl brine solution was injected evenly into the subsurface layers of the mesocosms on days 13 (69 L at a depth range of 10 – 17 m) and 33 (46 L at a depth range of 12.5 – 17 m) of the experiment. On day 14, Peruvian Scallop larvae (*Argopecten purpuratus*) were added in concentrations of ~10,000 individuals m⁻³, and on day 31 Fine Flounder eggs (*Paralichthys adspersus*) were added in concentrations of ~90 individuals m⁻³. However, few scallop larvae and no fish larvae were found in the mesocosms after the release indicating that their influence on the plankton community was likely small.

Samples of suspended organic matter for IPL analysis were collected from all 8 mesocosms at both the surface and subsurface sampling depths on non-consecutive days throughout the experiment. See details of changes in water depths above. Due to the labor- and time-intensive nature of IPL analysis, we focused on 4 mesocosms (two from each treatment) at both depths and



from 9 different days spanning the 50-day experiment. We filtered 5L of mesocosm water through pre-combusted, 142 mm
Advantec glass fiber filters (GF75142MM) of 0.3 μm pore size. All samples were wrapped in combusted aluminum foil and
shipped frozen to the Organic Geochemistry Laboratory at the University of Colorado, Boulder for IPL extraction and analysis.
**2.2 Water Column Physicochemistry**
Depth profiles of salinity, temperature, $O_2$ concentration, photosynthetically active radiation (PAR), and chlorophyll a (chl-a)
fluorescence were measured through vertical casts using the CTD60M sensor system (Sea & Sun Technology). $O_2$
concentrations were cross-verified with the Winkler $O_2$ titration method performed via a micro-Winkler titration method
described by Arístegui and Harrison (2002). Seawater $pH_T$ (pH on total scale), was determined spectrophotometrically using
m-cresol purple (mCP) indicator dye as described in Carter, et al. (2013); see Chen et al. (2021) for details. See Bach et al.
(2020) for additional detailed information of sampling methods for water column physicochemistry.
Samples for inorganic nutrients were filtered immediately upon arrival at the laboratories at the Instituto del Mar del Perú
(IMARPE) using 0.45 μm Sterivex filter (Merck). The subsequent analysis was carried out using a continuous flow analyzer
(QuAAtro AutoAnalyzer, SEAL Analytical) connected to a fluorescence detector (FP-2020, JASCO). The method for
analyzing $PO_4^{3-}$ followed the procedure outlined by Murphy and Riley (1962), while $Si(OH)_4$ was analyzed according to Mullin
and Riley (1955). $NO_3^-$ and $NO_2^-$ were quantified through the formation of a pink azo dye as established by Morris and Riley
(1963) and additional corrections to all colorimetric methods was achieved with the refractive index method developed by
Coverly et al. (2012). Ammonium concentrations were determined fluorometrically following the method of Kérouel and
Aminot (1997). Further methodological specifics and the respective limits of detection for each analysis can be found in Bach
et al. (2020).
**2.3 CHEMTAX Analysis**
Pigment samples were flash-frozen in liquid nitrogen directly after filtration and kept frozen on dry ice during transport to
Germany for extraction as described by Paul et al. (2015). Concentrations of extracted pigments were measured by reverse-
phase high-performance liquid chromatography (HPLC; Barlow et al., 1997) calibrated with commercially available standards.
The relative contribution of distinct phytoplankton taxa was calculated with CHEMTAX, a program for calculating the
taxonomic composition of phytoplankton populations (Mackey et al., 1996). Input pigment ratios specific to the Peruvian
upwelling system, determined by DiTullio et al. (2005) and further described by Meyer et al. (2017), were incorporated in
these calculations (see Bach et al., 2020).
**2.4 Flow Cytometry**
Samples (650 μL) from each mesocosm were analyzed using an Accuri C6 (BD Biosciences) flow cytometer. The signal
strength of the forward light scatter was used to distinguish phytoplankton groups, in addition to the light emission from red





fluorescence of chl-a, and the light emission from orange fluorescence of phycoerythrin. Size ranges were constrained via
gravity filtration using sequential polycarbonate filters ranging from 0.2 to 8 μm and the strength of the forward light scatter
signal. Additional details of this method can be found in Bach et al., 2017. In this study, we only report *Synechococcus* (0.8-3
μm) counts (cells mL$^{-1}$) because it is the only phytoplankton group that was consistently selected to exhibit statistically
significant trends with IPLs likely due to the non-size fractionated nature of IPL sampling.
**2.5 Lipid Extraction and Analysis**
Intact polar lipids were extracted from glass fiber filters via a modified version (Wörmer et al., 2013) of the original Bligh and
Dyer Extraction method (Bligh and Dyer, 1959). Samples were extracted by ultrasonication a total of five times, with three
different extraction mixtures. Two extractions were performed using Dichloromethane:Methanol:Phosphate buffer
(aq) [1:2:0.8, *v:v:v*], adjusted to a pH of ~7.4, followed by another two extractions using Dichloromethane / Methanol /
Trichloroacetic acid buffer(aq) [1:2:0.8, *v:v:v*], adjusted to a pH of ~2.0. A final extraction was performed with
Dichloromethane / Methanol [1:5, *v:v*]. After each addition, samples were vortexed for 30 s, sonicated for 10 min, and then
centrifuged for 10 min at 3000 rpm while kept at 10 °C. The supernatant of each extraction mixture was then transferred to a
separatory funnel where the organic fractions were washed and combined before solvent removal under a gentle N2 stream.
Before analysis, the TLEs were resuspended in dichloromethane:methanol (9:1 *v:v*) and filtered through a 0.45 μm
polytetrafluoroethylene (PTFE) syringe filter.

Chromatographic separation and identification of IPLs was achieved using a Thermo Scientific UltiMate 3000 high-
performance liquid chromatography (HPLC) interphase to a Q Exactive Focus Orbitrap-Quadrupole high resolution mass
spectrometer (HPLC-HRMS) via heated electrospray ionization (HESI) in positive mode, as described in detail in Cantarero
et al. (2020) and Flores et al. (2022). Samples were analyzed in full scan mode to obtain an untargeted screening (or lipidomic
profile) of each sample, in addition to targeted MS/MS mode for compound identification via diagnostic fragmentation patterns
(e.g., Sturt et al., 2004; Schubotz et al., 2009; Wakeham et al., 2012). IPLs were identified by their exact masses, polar head
groups, the total number of carbon atoms and unsaturations in the core structure, and their retention times. While other studies
have analyzed IPLs under both positive and negative ionization modes to determine the composition of individual fatty acid
chains in the core lipid structures, we took advantage of the high resolution of the Orbitrap mass spectrometer to focus on the
diversity of head group combinations with total carbon atoms and unsaturation only (Cantarero et al., 2020).

Quantification of IPLs was achieved using a combination of an internal standard added to samples during extraction, in addition
to an external calibration curve consisting of 17 standards representing different IPL classes (see Cantarero et al., 2020 for

00    details of all internal/external and deuterated standards). The intensity of each individual IPL identified in the HPLC-HESI-

01    HRMS analysis was calibrated to a linear regression between peak areas and known concentrations of the same lipid class (or

02    the most similar molecular structure) across a 5-point dilution series (0.001–2.5 ng/μl). Samples were analyzed across 3





separate analytical periods with weekly calibration curves to account for variation in the ionization efficiency of compounds over time. Overall, HPLC-ESI-HRMS is considered a semi-quantitative method due to changes in ionization efficiency of different IPL standards and environmental analytes. These changes are largely caused by differences in polar head group compared to the acyl chain length and degree of unsaturation (Yang and Han, 2011). Nonetheless, we investigate both relative (%) and absolute IPL abundances to compensate for the current analytical limitations in IPL quantification.

While we report IPL structural variations associated with different head groups and modifications in the core structure (i.e., unsaturation degree and carbon length of diacyl chains), we particularly focus on the former. Additionally, several IPLs thought to be absent in eukaryotic phytoplankton or far more abundant in bacteria and archaea (n = 34in total) have been removed to facilitate the analysis of trends that are predominantly phytoplanktonic in origin. These compounds include PME/PDME (Phosphatidyl(di)methylethanolamine) and intact GDGTs (glycerol dialkyl glycerol tetraethers) classes. In addition, while we lack detailed structural information of core individual fatty acids, their combined carbon atoms can be used to deduce short (i.e. < 14 carbon atoms per chain) or odd numbered fatty acid chains typically found in bacteria (Volkman et al., 1989; Volkman et al., 1998; Russel and Nichols, 1999; Jónasdóttir, 2019). Thus, compounds conventionally regarded as bacterial (34 in total) were also removed to minimize their impact on the analysis of predominantly phytoplanktonic IPLs (165 in total). We recognize that while this selection approach reduces the influence of non-eukaryotic lipids, we still cannot rule out an undetermined contribution from heterotrophic bacteria to the IPL pool in this experiment. A summary of these IPL classes and their acronyms is provided in Table 2.

## 2.6 Multiple Linear Regression

We performed multiple linear regression between 8 relevant environmental factors and 165 unique IPLs. With so many IPL-environmental factor pairs to analyze, we used multiple linear regression (MLR) for quick and easily digestible outputs. In each MLR model, the relative abundance of individual IPL molecules was employed as a response variable, with environmental factors serving as predictor variables. Additionally, phytoplankton abundances were included to account for their linear effects on IPL distributions. The rationale behind this inclusion lies in the understanding that variations in phytoplankton abundance may exert a proportional and predictable impact on the abundance patterns of specific IPLs.

We prioritized linear relationships with IPL relative abundances (% abundances) to emphasize changes in the proportions of phytoplankton lipids, rather than their absolute concentration and contribution to biomass. This approach enables us to distinguish compositional changes in the lipid pool from variability in total biomass production. Additionally, MLRs were constrained to focus on the most abundant IPLs in this system, defined as those constituting more than 2.5% to the total IPL





pool. This restriction was implemented to reduce noise associated with low-abundance IPLs, and to enhance the robustness of
the analysis.

We chose to investigate linear relationships within individual depths (surface and subsurface) to focus on temporal changes
within distinct environments, rather than comparing these two environments directly (see CART and Random Forest methods,
below). IPLs and environmental factors were permuted to tabulate the regression coefficient for each IPL-environmental factor
pair. Model coefficients were directly comparable due to centering and scaling of environmental and phytoplankton variables
(see eq. 1) to linearize the relationship and better align with the model assumptions:

$$Y = \beta_0 + \beta_1 X_1 + \beta_2 X_2 + ... + \beta_n X_n + \varepsilon \qquad \text{eq. (1)}$$
where, $Y$ is the dependent variable (or response), $X_1, X_2,...,X_n$ are the predictor variable (or independent variables), $\beta_0$ is the
intercept (constant term), $\beta_1, \beta_2,..., \beta_n$ are the coefficients representing the magnitude and direction of the relationship between
the predictor variable and the dependent variable, and $\varepsilon$ is the error term capturing the variability not explained by the model.

2.7 Classification and Regression Tree (CART) and Random Forest
Classification and Regression Trees (CART; Breiman et al., 1984) are predictive machine learning algorithms that partition
and fit data along a predictor axis into homogenous subsets of the dependent variable. Regression trees are used for dependent
variables with continuous values, while classification trees for categorical values. Here, we apply regression trees to diagnose
the environmental and biological variables that affect IPL concentrations by polar head group class. We limited the size of the
tree splits via a "pruning" process, where less significant variable splits are removed as determined by a deviance criterion
resulting in the best-fit tree with the least mean squared error. Additionally, predictor variables were selected in conjunction
with random forest analyses, which ranks the variable contribution to the model performance.

Random forests are derived from bootstrapping of observational data, and the generation of many decision trees based on each
bootstrap sample. For a covariate vector, the predicted estimate of IPL class concentrations is an average of the many randomly
configured decision trees with the same distribution. A randomly generated subset of decision trees is used to split each node,
enabling reduced variance and correlation amongst individual trees, and ideally improving the accuracy of model predictions
(Breiman, 2001). The random forest algorithm also allows for the ranking of predictor variables based on the prediction
performance and is reported here as the predictor's contribution in reducing the root mean square error (RMSE) in the model.
For additional details on the random forest algorithm, please see Hastie et al. (2009). All CHEMTAX, flow cytometry, and
physicochemical variables were included in the CART and random forest analyses as predictors. The model variables were



reduced via repeated random forest analyses to only include predictors that reduce the RMSE of the model predictions by at least 5%.

CART and random forest-based predictions serve as ideal methods to explore environmental drivers of IPL distributions- not only because of their predictive performance- but also because of their non-parametric nature, their diagnosis of variable importance, and their ability to handle non-linear interactions and small sample sizes (Tyralis et al., 2019). We do not employ these methods to predict the concentration of IPL classes, but rather to identify the primary environmental and biological drivers of change in IPL class concentrations, and potential interactions between environmental conditions and IPL remodeling amongst phytoplankton. Therefore, we focus our interpretations of both the CART and Random Forest analyses on only the most significant predictors and their relative order of importance in the model predictions.

## 3 Results

### 3.1 Oxygen and $pH_T$

Oxygen concentration in surface waters ranged between ~125 and 140 μmol/L before the ODZ water addition in all 4 mesocosms (Fig. 2A). The concentration dropped slightly (~15 μmol/L) following the water addition and steadily increased to a maximum between ~185-220 μmol/L by day 28. Day 30 marked a significant drop in oxygen concentration by 30-60 μmol/L in all mesocosms with conditions stabilizing between 140 and 155 μmol/L by day 50. The temporal variability in $pH_T$ in surface waters showed a largely mirrored trend to that of oxygen content. Before ODZ water addition $pH_T$ was 7.9-8.0, which dropped by ~0.2 immediately following the water addition. After a few days of relatively stable $pH_T$ at ~7.8 in all four mesocosms, the $pH_T$ increased significantly between days 18 and 28 reaching ~8.1-8.2. From days 30-40 the $pH_T$ gradually decreased by approximately 0.2 in all mesocosms until increasing again (most notably in mesocosms 5 and 7) between days 42-50 to maxima ranging from 8.1-8.3.

The subsurface waters were more oxygen-depleted compared to the surface and showed an earlier onset of increasing oxygen concentration, beginning immediately after the ODZ water addition. All four mesocosms reached maximum $O_2$ levels (60-75 μmol/L) by day 13 and began decreasing markedly by day 16. The lowest concentrations (~15 μmol/L $O_2$) were reached between days 30 and 34. Oxygen concentrations recovered slightly in the last 10-16 days of the experiment and were all within 30 (± 10) μmol/L of $O_2$. The subsurface waters, again, showed similar temporal patterns in $pH_T$ as with $O_2$, except for mesocosm 7. The $pH_T$ was variable (~7.60 ±0.05) in the early portion of the experiment (days 10-16) but did not show a dramatic response to the OMZ water addition as was the case in the surface samples. The $pH_T$ did gradually decrease from days 18-30, with $pH_T$ minima in all four mesocosms reached on day 30 (~7.45-7.50). Day 30 also marked the beginning of a significant increase in the $pH_T$ (~0.15-0.20) across all mesocosms with the $pH_T$ steadily increasing to ~7.65-7.70 by day 50. Additional details on the carbonate chemistry of the mesocosms can be found in Chen et al. (2021).





98

## 3.2 Nutrient Concentrations

In surface waters, the nutrients $NH_4^+$, $NO_3^-$, $NO_2^-$, $PO_4^{3-}$, and $Si(OH)_4$ all showed consistent trends over time, with the highest concentrations occurring either just before (day 10) or immediately after (day 12) the ODZ water addition (Fig. 2B). Nitrogen species ranged between 1 and 3 µmol/L during this early part of the experiment, while $Si(OH)_4$ ranged between 4 and 7 µmol/L, and $PO_4^{3-}$ remained at ~ 2 µmol/L. The concentration of all nitrogen species dropped quickly to near minimum values by day 15, and typically remained <0.5 µmol/L for the remainder of the experiment. $PO_4^{3-}$ remained replete (>1.5 µmol/L for the entirety of the experiment). $Si(OH)_4$ dropped to ~3-4 µmol/L by day 15 and gradually increased in all 4 mesocosms by 1-2 µmol/L until days 36-38, where concentrations gradually dropped to near minimum values of 3-4 µmol/L. There were periodic enrichments in $NH_4^+$, most notably between days 40-50, as discussed in Bach et al., 2020.

The subsurface-water nutrients showed similar temporal patterns as the surface but were generally more enriched in all nutrient species. All nitrogen species decreased in the few days following the ODZ water addition, but at a more gradual pace than in the surface. Before and immediately after the water addition, $NO_3^-$ was enriched at 4-6 µmol/L and $NH_4^+$ between 2.5-4 µmol/L. $NO_2^-$ concentrations were ~0.5-1 µmol/L in mesocosms 6 and 7 during this period but remained low (<0.3 µmol/L) in mesocosms 5 and 8. Broadly speaking, the nitrogen species all reached minimal values (<0.1 µmol/L) by days 18-20, except for in mesocosms 8 and 6, where $NO_3^-$ persisted at significant concentrations (0.2-2.3 µmol/L) until days 22 and 24, respectively. The remainder of the experiment was marked by typically depleted concentrations of all nitrogen species (<0.1 µmol/L) with occasional spikes in $NH_4^+$ reaching up to 1 µmol/L, particularly towards the end of the experiment. Subsurface $PO_4^{3-}$ concentrations were similar to the surface, remaining around ~2.0 µmol/L for most of the experiment which gradually decreased to concentrations of ~1.5 µmol/L by the end of the experiment. $Si(OH)_4$ similarly decreased from maximum concentrations of 4.0-4.5 µmol/L shortly after the ODZ water addition, and gradually decreased by ~ 1 µmol/L over the course of the experiment.

## 3.3 Total Chlorophyll a

Chl-a concentrations were highly variable throughout the experiment, particularly in subsurface waters (Fig. 2C). Concentrations were generally more elevated at the surface compared to the subsurface, with values ranging ~2.56-2.96 µg/L on day 10. In all mesocosms, the chl-a concentration increased to ~7.94-14.01 µg/L after the ODZ water addition and through day 20. Day 22 showed a significant drop in chl-a concentrations to between 1.35 and 2.89 µg/L in mesocosms 5, 7, 8. Chl-a concentrations at the surface remained rather constant until days 36-40, where concentrations rapidly increased until maximum concentrations on days 48-50 (14.0-47.3 µg/L). In subsurface waters, chl-a concentrations were notably lower than in the surface and ranged between 0.58-0.84 µg/L on day 10. After the ODZ water addition, chl-a concentrations increased slightly until day 14, but did not show consistent distributions amongst the 4 mesocosms afterwards; decreases were observed in





mesocosms 5, 6, and 8, but an increase in mesocosm 7. Chl-a concentrations increased rapidly on day 22 in all 4 mesocosms ranging between 4.03 and 8.44 µg/L. While remaining highly variable, the concentrations generally increased after day 24 with a notably abrupt increase between days 40-44 to near maximum values ranging between 2.58-11.3 µg/L.

**3.4 Phytoplankton Community Composition**

The CHEMTAX based phytoplankton community compositions demonstrated more variability in phytoplankton assemblages in the subsurface samples than in the surface (Fig. 2C). Before the ODZ water addition on day 10, surface waters in all 4 mesocosms showed similar phytoplankton distributions with high relative abundances of Bacillariophyceae, referred to from here on as diatoms (20-45%), Chlorophyceae (15-50%), and Dinophyceae, referred to as dinoflagellates (25-45%). These distributions remained rather similar immediately after the water addition (day 12), but dinoflagellate contributions increased in the following days and ranged from ~20% of the total chl-a pool to up to 75% by day 20. Of note, Cryptophyceae made minor contributions to the chl-a pool aside from days 15-18 in mesocosm 6 where they contributed up to 25%. Dinoflagellates largely dominated the chl-a pool for the remainder of the experiment with moderate blooms of diatoms between days 34-44 in mesocosms 7 and 8.

Subsurface waters exhibited greater variability in the phytoplankton assemblages and a greater contribution of Chlorophyceae, Cryptophyceae, *Synechoccocus*, and diatoms to the chl-a pool than in the surface. Pre-addition waters (day 10) were dominated by diatoms making up >75% of the chl-a pool. In the first few days after the ODZ water addition (days 12-15), the relative abundance of Chlorophyceae increased to 25-45% in mesocosms 5, 6 and 8, whereas Cryptophyceae contributed between 5-15% of the total chl-a. During this period, diatoms continued to dominate mesocosm 7 but decreased gradually from 65 to 15% of the phytoplankton community. Notably, in mesocosm 6, Cryptophyceae contributed a moderate amount to the chl-a pool beginning on day 15 (25%) and gradually increased to 40% by day 20. Mesocosms 5, 7, and 8 increased in Cryptophyceae as well, but were limited to ~10-25%. Similar to surface waters, dinoflagellates dominated the chl-a pool by day 20 and contributed 50-80% of the chl-a pool; however, there was considerably greater variability in dinoflagellate abundance after day 20 in subsurface waters. Chlorophyceae remained a significant contributor for the rest of the experiment typically ranging between 10 and 25% of the phytoplankton relative abundance. In mesocosms 5, 7, and 8 diatoms became the dominant contributor, totaling ~50% of the phytoplankton between days 32 and 44. By the final days of the experiment (days 48-50), dinoflagellates again made up most of the phytoplankton community (60-75%). Pelagophyceae, Prymnesiophyceae, and Cyanophyceae (referred to here as *Synechococcus*), remained minor contributors throughout the entire experiment but showed maximum contributions of <10% in days 10-16. Mesocosm 6 showed a minor contribution (<10%) of *Synechococcus* from days 42-50, and from days 34-42 in mesocosm 7.





### 3.5 IPL Class Distributions

The IPL distributions throughout the study can be summarized by the relative abundances of different classes determined by their polar head groups. IPL distributions were broadly consistent between mesocosms and treatments during the experiment but showed significant differences between surface and subsurface waters (Fig. 2D). Glycolipids such as SQ, DG, and MG typically made up ~50-75% of the total IPL pool in the surface, whereas phospholipids such as PC and PG were dominant in the subsurface (~50-75%). The changes in IPL distribution from the surface samples over the course of the experiment were most apparent between days 10 to 12, 20 to 24, and 24 to 38. Day 10 of the experiment marks the sampling period immediately preceding the ODZ water addition and was the only day with a significant fraction of BLs (betaine lipids) in the total lipid pool (ranging from 25-30% in the surface samples). Mesocosms 6 and 8 showed a similar distribution between days 10 and 12 of the experiment with large contributions of SQs (~50%), BLs (~25-30%), and MGs (~20%), while mesocosms 5 and 7 showed considerably more PCs (~25-40%) at the expense of SQs (<10%). In the 2 weeks following the deep-water addition (days 12 to 24), we saw increasing (albeit variable) relative abundances of PGs (from 5 to 50%) and DGs (from 5 to 25%) in surface waters. MGs made up a considerably larger fraction during this time in mesocosm 6 (up to 50%), but were more moderate in mesocosms 5, 7, and 8 (typically ~25%), yet decreased in all mesocosms to <5% by day 24. The final sample days (38 and 50) showed a resurgence of glycolipid contributions, namely in SQs (25-55%) and MGs (5-25%), with moderate contributions of PGs (5-20%) and DGs (5-20%).

In the subsurface samples before the ODZ water addition, mesocosms 6, 7, and 8 showed moderate contributions of MGs (15-25%) which was lower in mesocosm 5 (<5%). There were higher contributions of SQs in mesocosms 6 and 8 (50% and 35%, respectively) than in 5 and 7 (20-25%). Instead, mesocosms 5 and 7 showed a greater contribution of PCs (45-55%). All 4 mesocosms demonstrated some component of PGs ranging from 5-20% of the total IPL pool. The 2 weeks following the ODZ water addition showed highly variable fluctuations between PCs and PGs as the dominant IPL classes, with the contributions of glycolipids (25-40%) and betaine lipids (<5%) remaining consistent. Of note, day 24 marked a consistently low contribution of PCs, which persisted until the end of the experiment. The final sample days (38 and 50) showed similar contributions between MGs, SQs, and PGs that together dominated the total lipid pool.

### 3.6 Multiple Linear Regressions

We found statistically significant ($p < 0.05$) linear relationships between the relative abundance of individual IPLs and environmental factors (Figure 3). In surface waters, $pH_T$ showed several significant responses towards DGs, MGs, PGs and SQs. Amongst DG, MG, and SQ, there were both positive and negative relationships with $pH_T$; IPLs containing 3 or more unsaturations showed negative relationships, and molecules with fewer than 3 unsaturations showed positive relationships. PGs, however, showed exclusively negative linear responses to $pH_T$. In the subsurface, $pH_T$ only imparted significant linear effects towards SQs with a similar pattern of generally more unsaturations positively related to $pH_T$.





The temperature of surface waters showed predominantly positive regression coefficients with several PG and MG molecules, with inconsistent correlations found in DG and SQ. Whereas the remaining IPL classes showed significant linear relationships with temperature, their regression coefficients were near 0. Similarly, in the subsurface only 2 IPL structures (DG and SQ) showed a significant linear response but with regression coefficients near 0.

Oxygen concentrations showed few linear correlations at the surface and were limited to two BLs (negative) and 1 PG (positive). In the subsurface, the strongest linear correlations were found between oxygen and PGs (negative) and SQs (mostly negative). Whereas other linear relationships did appear with oxygen, they were comprised of near 0 regression coefficients.

Nutrient concentrations showed many significant linear responses with regression coefficients of higher magnitude (up to $\pm 8$) and adjusted $R^2$ values of up to 0.5. The concentration of various forms of inorganic nitrogen had largely positive linear responses to BLs and PCs in both the surface and subsurface, with strongly negative responses in PGs and MGs (in the subsurface). Among the SQs, many individual molecular species, with different fatty acid chains, responded linearly to all forms of inorganic N- however, the signs of these relationships were inconsistent in the surface, and generally positive at depth (aside from $NO_2^-$). DGs showed few linear responses and only in the subsurface, with similarly inconsistent signs. PE-IPLs showed little response to inorganic nitrogen concentrations. $PO_4^{3-}$ concentrations did not show many significant responses aside from BLs (negative) and PGs (slightly positive) in the surface. However, in the subsurface, $PO_4^{3-}$ showed strong linear responses to several MG and PG molecules, and strong positive relationships with PCs.

Light showed few linear responses amongst IPL relative abundances. At the surface, the strongest relationships were amongst PG (positive) and SQ (mixed signs) and to a slight degree BLs. In the subsurface, few significant relationships were noted and none with significant regression coefficients, indicating a range of light saturation with little to no linear effect on IPL distributions.

**3.7 CART and Random Forest**

All the predictive tree-based models showed improved performance with the inclusion of environmental variables (Figs. 4-7). The CART decision trees iteratively identified the key biological and physicochemical variables producing the best performing model in the prediction of IPL concentrations. The random forest analysis compliments these best fit decision trees by calculating the % reduction in the root mean square error (RMSE) associated with each variable. In conjunction, these two analyses highlight the most impactful variables in predicting the concentrations of a given IPL class. Overall, model performance amongst each IPL class can be compared by the strength of the correlation coefficients between observed and predicted concentrations, and by the magnitude of the RMSE (Figs. 4-7).





Amongst several IPL classes (i.e., SQs, DGs, and BLs), $pH_T$ was consistently a significant contributing variable to IPL concentrations, as demonstrated by both the CART and random forest analyses (Fig. 4A-D, and Fig. 5). Notably, $pH_T$ was identified as the most important variable amongst the best fit decision trees for both SQs and DGs, as well as the random forest model for SQs. Oxygen concentration was also frequently identified as a major contributing variable to model performance with MGs, DGs, PEs, PCs, (Figs. 4E-H, 5A-D, 6E-H, and 7A-D, respectively) and moderately important in SQs (Fig. 4A-D). Temperature was selected as the most important variable in PE predictions (Fig. 6E-H), and a major contributing variable in SQ, MG, BL, PG, and PC predictions (Figs. 4-7). Various forms of biologically available nitrogen were also important in the prediction of MG (Figs. 4E-H), BLs ($NH_4^+$; Figs. 5E-H), PE ($NH_4^+$; Figs. 6E-H), and PG ($NH_4^+$ and $NO_2^-$; Figs. 6A-D). $PO_4^{3-}$ concentrations showed significant contribution to model performance amongst BLs (Figs. 5E-H, denoted as Si:P ratios), SQs, MGs, PGs, and PEs (Figs. 4A-D, 4E-H, 6A-D, and 6E-H, respectively). Finally, light availability demonstrated secondary but significant importance in the prediction of SQs, MGs, PGs, PEs, and PCs (Figs., 4A-D, 4E-H, 6A-D, 6E-H, and 7 respectively).

Variables indicative of biological abundance were also identified as highly impactful to model performance, with chl-a concentration showing a significant contribution to all lipid classes except BLs (Figs. 4-7). Individual phytoplankton abundances also showed to be important predictive variables such as Cryptophyceae abundances for BLs, dinoflagellate abundances for DGs and SQs, *Synechococcus* abundances for DGs, PCs, PEs, and PGs, and diatoms for PCs.

**3.8 Water Treatments**

The experiment consisted of applying two different treatments to the mesocosms, aimed at exploring the varying impacts of upwelling of ODZ waters with contrasting geochemical properties. The first treatment saw the introduction of water from a coastal area (station 1) with ODZ waters with a very low N:P ratio (0.1; Table 1) into the mesocosm. The second treatment performed the same process, but with ODZ water from an offshore area (station 3) with a low N:P ratio (1.7; Table 1). Despite the different chemical signatures of both water-masses, the resulting nutrient stoichiometries within the mesocosms were similar in between both treatments (see Bach et al., 2020). We see largely similar responses in the IPL distributions, as well as in other biogeochemical variables between these two treatments; therefore, we largely focus our discussion on the temporal variation and differences between surface and subsurface environments in these analyses.

**4 Discussion**

**4.1 Biological Abundances as Drivers of IPL Distributions**

We expect phytoplankton abundances to exert first order control on the distribution of IPLs, as these molecules have been demonstrated to be chemotaxonomic biomarkers (Sturt et al., 2004, Schubotz et al., 2009, Wakeham et al., 2012; Van Mooy and Fredricks, 2010; Cantarero et al., 2020). Indeed, most IPL classes demonstrate phytoplankton abundances as a primary or





major predictor in the CART and random forest analyses, such as dinoflagellates in the prediction of SQs (Figs. 4A, C), DGs
(Figs. 5A, C), and PE (Fig. 6G), *Synechococcus* in the prediction of PGs (Fig. 6A, C), PEs (Fig. 6E, G), and PCs (Fig. 7A, C),
and chl-a in the prediction of every IPL class barring BLs (a mostly minor IPL class in this experiment). Chl-a appears most
important in highly abundant IPL classes as an indicator of overall photosynthetic productivity, and dinoflagellates dominate
the overall phytoplankton biomass for almost the entirety of the experiment post ODZ water addition. *Synechococcus*
demonstrates covariability with the total phytoplankton biomass, thus its perceived importance in controlling the abundance
of several IPL classes is likely due this covariance, as *Synechococcus* remains a relatively minor phytoplankton class. We
suggest that the prevalence of biological sources as IPL predictors in the decision tree analyses generally indicates the
variability in total phytoplankton biomass throughout the experiment.
Glycolipids MG, DG, and SQ are predominantly found in thylakoid membranes, whereas phospholipids PC and PE, as well
as BLs are structural components in the cell membrane lipid bilayer (note PG is found in both; Guschina and Harwood, 2013).
It is important to recognize that the relative proportions of IPLs vary in different phytoplankton classes (Harwood and Jones,
1989; Wada and Murata, 2009). Thus, the overall community composition, as well as the relative abundance of each species
within the phytoplankton community, are expected to be major drivers of IPL distributions. Apart from a minor contribution
from cyanobacteria (Fig. 2C) to the chl-a pool (presumably the IPL pool as well), our dataset primarily focuses on
photosynthetic eukaryotes. Accordingly, we recognize that phytoplankton abundances and community composition play a
major role in the distribution of IPLs. Thus, while we may refine the specificity of biological sources of IPLs in marine systems,
our aim is to explore the role of environmental forcing as an additional control on IPL distributions. The following sections
focus on the evidence for direct environmental influence on IPL remodeling amongst the phytoplankton community and the
potential implications of these physiological responses to the broader biogeochemistry.

## 4.2 Environmental Variables as Drivers of IPL Remodeling

In many cases, community composition changes concurrently and/or in response to environmental conditions. Therefore, we
employed two distinct strategies to isolate the role of lipid remodeling as a physiological response to environmental forcing
only. Firstly, the MLRs which subtract the variability explained by phytoplankton abundance (e.g., CHEMTAX results) in
pairwise correlations between lipid abundances and environmental variables. Secondly, the decision tree analyses (CART and
Random Forest) which rank variables by their impacts on the model performance.
Across virtually every major IPL class common to eukaryotic phytoplankton we see evidence of environmental conditions
forcing significant control on IPL polar headgroup distributions in the MLRs (Fig. 3). Similarly, physicochemical variables
such as nutrient concentrations, $pH_T$, temperature, $O_2$ concentration, and light availability are consistently identified as
statistically important variables in the prediction of IPL head groups in both the CART and Random Forest analyses (Figs. 4-



7). In addition, high level comparisons in the relative abundances of IPLs and phytoplankton groups (Figs. 3C, D) suggest that certain environmental conditions may be associated with major shifts in IPL distributions.

An important distinction between the MLRs and the decision trees is that the MLRs are calculated within individual depth environments (surface and subsurface) to explore statistically significant linear relationships between abundant IPL molecules and changing environmental conditions. On the other hand, the decision trees explore the predictive power of physicochemical differences between the surface and subsurface environments on IPL distributions. The results of these two analyses are meant to be complimentary, in that they focus on the differences in environmental conditions between water depths as well as the temporal development of conditions within a given depth over the course of the experiment.

### 4.2.1 Nutrient Availability

Nutrient limitation amongst phytoplankton leads to transitions in cellular activity, from the biosynthesis of growth/reproduction cellular components such as cell membranes to energy storing molecules (Guschina and Harwood, 2013, Zienkiewicz et al., 2016). Nitrogen, an essential nutrient in photosynthesis and the biosynthesis of proteins/enzymes and nucleic acids, is typically acquired by marine phytoplankton through inorganic nitrogen species such as $NO_3^-$ and $NH_4^+$. Some phytoplankton can also utilize organic nitrogen sources (Bronk et al., 2007), whereas diazotrophic cyanobacteria are able to fix dinitrogen gas into bioavailable nitrogen. The coastal region of the ETSP is typically considered to be seasonally co-limited by light, N, and Fe (Messié and Chavez, 2015). However, along the Peruvian shelf Fe concentrations are elevated compared to offshore waters (Hutchins et al., 2002; Browning et al., 2018), and is not considered a limiting source in this mesocosm experiment (Bach et al., 2020). The inorganic N:P ratio ranged between 0.13 and 4.67, with relatively enriched inorganic N in subsurface waters compared to the surface, and with a N:P minimum reached by day 20. Thus, we consider this mesocosm system to be overall nitrogen limited with varying degrees of severity. This is reflected in the transition from predominantly diatoms, Chlorophyceae, and Cryptophyceae to a dominance of mixotrophic dinoflagellates approximately 4-6 days after the initial ODZ water treatment (Bach et al., 2020). Such shifts are consistent with ecological changes from bloomer to survivalist species associated with cellular resource reallocation (Arrigo, 2005).

Under P limitation, phytoplankton are known to substitute non-phosphorus-containing glycolipids for phospholipids and reallocate the liberated P for other cellular demands (Van Mooy et al., 2009). More generally, nutrient limitation can cause phytoplankton to accumulate highly concentrated stores of energy in the form of triacylglycerols (TAGs) through the activation of multiple biosynthetic pathways (Zienkiewicz et al., 2016). These include synthesis via enzymes associated with the endoplasmic reticulum (Liu and Benning, 2013), de novo TAG synthesis in the chloroplast (Fan et al., 2011), as well as from phospholipids acting as donors of acyl units for TAGs production via the PDAT (phospholipid:diacylglyceroltransferase) enzyme (Dahlqvist et al., 2000). The PDAT enzyme can be a significant pathway for TAG accumulation (Popko et al., 2016;





Gu et al., 2021) and the encoding genes have been identified so far in green algae, diatoms, and heterokonts (Zienkiewicz et
al. 2016).
In our study, all four mesocosms experienced limitations in inorganic nitrogen (ranging from 0.24 and 4.29 µmol/L), and
consistently high concentrations of $PO_4^{3-}$ (ranging from 1.3 to 2.3 µmol/L) throughout the entire experiment. While random
forest analysis only detected a significant impact of inorganic N concentration amongst BLs, PGs, and MGs (Figs. 5G, 6A,
and 4E, respectively), the MLRs also suggest several individual molecules with significant linear relationships amongst PCs
and SQs (Fig. 3). Of note, the distributions of several abundant PCs and BLs with N in the headgroup composition are
positively correlated with inorganic N species. Other IPLs associated with N availability (PG, SQ, and MG) are generally
negatively associated, meaning that they are more abundant under more severe N-limitation.
Amongst *Chlamydomonas reinhardtii*, Cr-PDAT (a homologous enzyme to PDAT) appears to function as both an
acyltransferase and lipase, with the potential to use chloroplast membrane lipids as substrates to directly accumulate TAGs or
produce them *in vivo* (Yoon et al., 2012). In that study, Cr-PDAT showed the strongest lipase activity towards MGs, SQs, and
PGs under N-replete conditions and in logarithmic growth phase. Such a relationship could potentially explain the greater
relative abundances of these IPL classes in surface waters of our experiment (Fig. 2B) and/or negative linear regressions with
respect to N concentration at depth where inorganic N is more available (Fig. 3). The acyltransferase activity of Cr-PDAT,
however, has only been demonstrated in MG, and the overall role of Cr-PDAT in TAG accumulation was less pronounced
after prolonged N-deficiency (Yoon et al., 2012). Nonetheless, a mutant *Chlamydomonas reinhardtii* cultured with a disrupted
galactoglycerolipid lipase-encoding gene (known as plastid galactoglycerolipid degradation; PGD1), showed reduced TAG
production under N deprivation, suggesting that some MGs could serve as important substrates for fatty acids esterified into
TAGs (Li et al., 2012). However, these authors observed little net change in the membrane lipid relative abundances,
presumably as a way for this organism to maintain essential photosynthetic function, which suggests a maximum threshold of
membrane lipid recycling under N-limited conditions.
We suggest that N-bearing IPLs such as PC and BL may be ideal membrane lipids for PDAT-mediated degradation to serve
as both a source of acyl units for TAG accumulation and a mechanism for alleviating cellular N demand in low inorganic N
conditions. Under high growth rates, however, MGs, SQs, and PGs may serve as important substrates for TAG production.
Both PC and BL are extra-chloroplast membranes (Kumari et al., 2013), and thus non-essential to the photosynthetic
machinery, hence PC and BL may be favored for the repurposing of acyl units in surface waters where N availability is most
limited. Other environmental conditions (see sections below) may also influence the relative proportion of predominantly
thylakoid membranes IPLs (PG, SQ, MG, and DG) with essential functions in photosynthesis. Furthermore, the average ratio
of total IPLs/chl-a is up to three times higher at depth than at the surface (Fig. S5), demonstrating that the proportion of





membrane lipids in phytoplanktonic biomass at the surface is reduced and likely a result of known environmental stressors
inducing TAG production (i.e., nutrient limitation, oxygen stress, temperature, and light saturation).

**4.2.2 $pH_T$ and Inorganic Carbon Availability**

Coastal upwelling zones are characterized by low $pH_T$ subsurface waters associated with ODZs, where high fluxes of organic
substrates sustain enhanced microbial respiration and the accumulation of $CO_2$ (Capone and Hutchins, 2013). Thus, we explore
evidence for membrane lipid remodeling amongst phytoplankton as a physiological response to varying $pH_T$. We see evidence
of $pH_T$ imparting a control on the composition of IPL head groups, particularly amongst SQs and DGs, as noted by the high
importance rankings in both the CART and Random Forest analyses (Figs. 4A,C and 5A,C). This likely represents the
relatively high abundance of these glycolipids in the surface samples where the $pH_T$ is 0.2-0.6 higher than at depth. The MLRs
show a complex relationship between individual IPL moieties and $pH_T$, with both negative and positive linear regressions of
statistical significance ($p < 0.05$; Figs 3-4). The increased proportion of unsaturated IPLs associated with lower $pH_T$
(particularly amongst DG, MG, and to some extent SQ) is most apparent in surface waters where the variability in $pH_T$ is
greatest ($\pm 0.2$). It has been suggested that lower $pH_T$ can induce production of saturated fatty acids as a mechanism to reduce
membrane fluidity and prevent high proton concentrations in the cytoplasm (Tatzusawa et al., 1996); however, only limited
responses have been observed in considerably more extreme $pH_T$ ranges (e.g., ~1-10) amongst *Chlamydomonas sp.*
(Poerschmann et al., 2004).

Rather than the direct effects of modest changes in $pH_T$ on algal membrane fluidity, the observed changes in fatty acid profiles
may be a result of other carbonate chemistry parameters in this mesocosm system. The surface waters after the ODZ water
addition showed rapid declines in DIC and $pCO_2$ within a few days (Chen et al., 2021) due to high productivity, as is typical
of the ETSP with $pCO_2$ maxima occurring in the ODZ where respiration rates are high (Vargas et al., 2021). Lower $pCO_2$
concentrations may be a limiting factor for photosynthesis and growth; higher $pH_T$ in marine settings indicates a reliance on
active transport of $HCO_3^-$ for carbon fixation as opposed to a passive diffusion of $CO_2$ (Azov et al., 1982; Moazami-Goudarzi
et al., 2012). Enrichment of $pCO_2$ has also been observed to induce an increased proportion of unsaturated fatty acids in
microalgae (Morales et al., 2021), which may explain the negative correlation between the proportion of unsaturated
glycolipids (as well as PGs) and pH (high $pCO_2$; see Fig. 3). It has been suggested that elevated $CO_2$ levels may facilitate an
increase in photosynthetic rates and the provision of more acetyl units (Hu and Gao et al., 2004).

Another possible explanation for the association between IPLs and $pH_T$ is a stressor response in the form of TAG production.
As discussed above, PDAT, and the homolog Cr-PDAT, enable phytoplankton to repurpose phospholipids and galactolipids
as substrates for TAG production. Notably, Yoon et al. (2012) were able to demonstrate the suitability of MG as a substrate,
but not DG or SQ. This may partially explain the higher proportion of SQs and DGs overall under higher $pH_T$ (where $pCO_2$ is
lowest) and when growth rates have slowed following initial bloom conditions. While nutrient limitation is often considered





the primary regulator of TAG production, culture experiments of TAG production in chlorophyte CHLOR1 under alkaline $pH_T$ stress led to cellular TAG accumulation, as well as a reduction of membrane lipids (Guckert and Cooksey, 1990). Additionally, the chlorophyte *Scenedesmus sp.* WC-1, when analyzed in a $pH_T$ buffered system under differing $NO_3^-$ concentrations, demonstrated that both higher pH and N-limitation increased TAG production individually. The synergistic impact of both higher pH and N-limitation resulted in a notable enhancement of TAG production (Gardner et al. 2011).

Notably, enhanced TAG production was not observed in cultures of the diatom *Phaeodactylum tricornutum* with bicarbonate amendments (Gardner et al., 2012). Other experiments with *Phaeodactylum tricornutum* demonstrated a significant reduction in total membrane lipid content and enhanced TAG synthesis when the diatom experienced nutrient limitation but was provided with abundant $CO_2$. This result indicated that TAG production in this model diatom is regulated by nutrient limitation (either N or P), but carbon availability may promote it by providing substrates for TAG synthesis (Peng et al., 2014). It remains unclear why TAG production differs in relation to $pH_T$ amongst these selected green and diatom algae, but differences in carbon concentrating mechanisms may be involved (Gardner et al., 2012), implying a varied flexibility in phytoplankton to uptake alternate forms of inorganic carbon (i.e., $HCO_3^-$). Again, the generally higher ratios of total IPLs/chl-a in subsurface waters compared to the surface (Fig. S5) indicates a greater proportion of membrane lipids in phytoplankton biomass. These ratios suggest that TAG production is more active in the surface, which may be in part related to the active transport of $HCO_3^-$ in phytoplankton cells for photosynthesis.

Differences in the effect of carbonate chemistry on TAG synthesis amongst phytoplankton classes may have implications on community composition under changing environmental conditions. The ability of phytoplankton to adapt to environmental stressors via TAG production may relegate them to certain depths of the water column based on the combined effects of nutrient availability and $pH_T$ (amongst other variables, e.g., light, temperature, $O_2$). Based on the dominance of dinoflagellates in surface waters once the inorganic N is consumed (Fig. 2B and C), it is possible that this phytoplankton class may be actively recycling membrane lipids to produce energy storing TAGs. This may represent one of several physiological advantages of dinoflagellates over diatoms in N-limited conditions, as the latter have not shown enhanced TAG synthesis at higher pH in culture (Gardner et al., 2012).

Overall, our results agree with other experimental data in that pH may play a significant role in the distribution of IPL head groups amongst certain phytoplankton groups. The relatively high glycolipid abundances in surface samples with higher $pH_T$ (most notably SQ and DG, see Fig. S1) are likely not related to direct effects on membrane fluidity, but rather to the availability of $CO_2$ for TAG synthesis and the concomitant recycling of membrane lipids. Additional analyses of TAG concentration amongst the phytoplanktonic community and their fatty acid compositions would aid in tracing the extent of membrane lipid degradation as a source of acyl units, as well as to trace the location of these biosynthetic pathways in the cell.



### 4.2.3 Oxygen

Despite the harvesting of light energy during photosynthesis, photosynthetic organisms also rely on respiration for growth and free radical scavenging under both light and dark conditions (Raven and Beardall, 2003). Specifically, the availability of $O_2$ (particularly during the dark phases or in low-light environments) influences the activation of metabolic responses, such as fermentative metabolism and acetate utilization (Yang et al., 2015). Differences in dark respiration rates relative to light-saturated photosynthesis among algae may confer advantages under varying oxygen availability (Geider and Osborne, 1989). Thus, the capability of some organisms to perform lipid remodeling in response to oxygen stress may play a role in shaping the composition of the phytoplankton community.

In our analysis, the Random Forest models indicate a significant impact of $O_2$ concentration in the prediction of nearly all IPL class abundances (Figs. 4-7). This pattern appears to be largely driven by differences in IPL distributions between surface (oxygenated) and subsurface (hypoxic; $< \sim 1.4$ ml/L as defined by Naqvi et al., 2010) waters. Glycolipids (MG, DG, and SQ) make up on average 28% more of the IPL pool in surface waters ($\sim$125-220 µmol/L) compared to subsurface waters ($\sim$15-75 µmol/L). The MLRs, however, show no significant relationships between glycolipid relative abundances and $O_2$ concentration in surface samples (Fig. 3); we posit that this is due to surface waters remaining well oxygenated throughout the experiment. The deep samples indicate few statistically significant linear relationships, most notably a negative relationship with several PG molecules and an inconsistent relationship amongst SQs (Fig. 3). This suggests that the most prominent relationship is driven by a change from oxic to hypoxic conditions (i.e., surface vs subsurface) and not a sensitivity to increasingly depleted $O_2$ concentrations.

Oxygen availability appears to impart a significant control on the concentration of MG and DG glycolipids, as evidenced by their high relative abundances in the well oxygenated surface waters and their high importance in the Random Forest models. We interpret this signal to potentially indicate remodeling of these thylakoid lipids in response to oxygen stress. In subsurface waters, the relative abundance of other thylakoid lipids, such as PG and SQ, do show generally negative linear relationships with $O_2$ concentration. We suggest that this indicates a gradual transition of IPL distributions between oxic and hypoxic environments.

Anaerobiosis amongst green algae has been demonstrated to impact lipid production, with significant reductions (by nearly 50%) in polar lipid content and concomitant increases in fatty acids (Singh and Kumar, 1992). Gombos and Murata (1991) found that the cyanobacterium *Prochlorothrix hollandica* experienced a significant reduction in the relative abundance of MGs that coincided with moderate increases in SQs, DGs, and PGs under low oxygen conditions. Furthermore, culture experiments of anaerobically grown *Chlamydomonas reinhardtii* resulted in both decreased membrane lipid yields (most notably amongst MGDGs and DGDGs; by > 50%) and an accumulation of TAGs (Hemschemeier et al., 2013). It has been noted that oxygen stress appears to induce the degradation of fatty acids ($\sim$30% reduction under dark/anaerobic conditions), mostly amongst





unsaturated fatty acids commonly found in MGDG and DGDG membrane lipids (16:4 and 18:3) used for TAG assembly (Liu,
2014; Hemschemeier et al., 2013). Glycolipids (i.e., MG and DG) appear to serve as important substrates for TAG production
under low oxygen conditions; however, beta-oxidation of fatty acids requires oxygen to contribute to the degradation of acyl
groups, potentially explaining why membrane lipid degradation is attenuated under more severe hypoxia (Liu, 2014). This
physiological response to low oxygen conditions in subsurface waters may explain the relatively high abundance of glycolipids
in well oxygenated surface waters.
Dinoflagellates have been shown to exhibit particularly high dark respiration to light-saturated photosynthetic rates as
compared to diatoms, Chlorophyceae, or most notably cyanobacteria (Geider and Osborne, 1989), possibly pointing towards
a greater sensitivity to $O_2$ concentrations. In addition to other environmental conditions, the greater relative abundance of
Chlorophyceae, diatoms, and Cryptophyceae in the oxygen-deficient subsurface waters may reflect reduced respiration rates
amongst these algae. Differences in the proportion of glycolipids MG and DG amongst different algae, and their relative ability
to recycle them under oxygen stress, may play a prominent role in their individual tolerances to oxygen limitation.
Higher proportions of glycolipids in surface waters may also be due to enhanced rates of microbial degradation under oxic
conditions, which may be 2-4 times faster than under anoxic conditions, as tested in microcosm experiments (Ding and Sun,
2005). Relatively labile glycolipids can accumulate in the dissolved organic carbon pool (Gašparović et al., 2013). This
observation aligns with the slower breakdown of SQs compared to phospholipids observed in IPL degradation experiments
under aerobic conditions (Brandsma, 2011). This accumulation process, however, is unlikely in regions of the water column
with large numbers of active living cells, and highly active bacterial degradation. In fact, the distributions of IPLs across the
ODZ of the ETSP indicate minor contributions of exported IPLs to greater depths, suggesting high surface recycling (Cantarero
et al., 2020). However, specific experimental observations encompassing oxygen gradients ranging from well-oxygenated to
fully anoxic conditions are necessary to derive more robust conclusions.

### 4.2.4 Temperature

Phytoplankton have been shown to respond variably to high growth temperatures depending on their individual tolerances
(Huertas et al., 2011). Photosynthesis is considered the most heat-sensitive cellular function in photoautotrophs (Berry and
Björkman, 1980). In this section we discuss the potential lipidomic responses to heat stress within the IPL distributions of the
phytoplanktonic community. Temperature fluctuations affect membrane fluidity, a phenomenon commonly controlled by fatty
acid desaturases (Sakamoto and Murata 2002) that catalyze the production of unsaturated/saturated fatty acids to
increase/decrease membrane fluidity, respectively (Lyon and Mock 2014). We did not see evidence for temperature effects on
the degree of unsaturation in our experiment (see Fig. S2).





This is likely due to the narrow temperature range observed during the experiment (17.3 - 21.6 ºC), which mirrors the natural variability observed in Callao (average monthly ranges ~16.6-19.6 ºC from 2017 to 2019(Masuda et al., 2023). Despite the restricted temperature range and its lack of impact on the unsaturation degree of core lipids, the random forest analyses identified temperature as a significant variable in predicting all IPL classes based on their polar head groups aside from DG and PE (Figs. 4-7). This suggests that the response could vary among different IPLs classes The MLRs indicate consistently positive relationships between several glycolipids and temperature, suggesting a potential physiological compensation via membrane compositions for higher temperatures. Gašparović et al. (2013) noted an accumulation of glycolipids at temperatures >19 ºC in the northern Adriatic Sea, particularly from cyanobacterial synthesis of MGs. The sensitivity of photoautotrophs to thermal stress was also explored by Yang et al. (2006), who showed that DGs and MGs both increase the thermal stability of photosystem II, while phospholipids significantly decrease it. Experiments with a wild-type and mutant *Chlamydomonas reinhardtii* have shown that SQs are an essential component of thylakoid membranes to maintain stability under heat stress (Sato et al., 2003), although at considerably more extreme temperatures (41 ºC). Heat stress has also been linked to the production of TAGs (Elsayed et al., 2017; Fakhry and El Maghraby, 2015), which can draw acyl units from degraded membrane lipids (Holm et al., 2022).

Interestingly, the relative abundance of DGs in our experiment shows the most prominent ($R^2$ of 0.35-0.44), and statistically significant ($p < 0.05$), linear relationship with temperature (see Fig. S3A). While temperature was not identified as an important variable to the prediction of DGs in either decision tree analysis, this may be due to other covariates, such as pH and $O_2$, masking the effect of temperature. Our results indicate that phytoplankton may either produce DGs in greater abundance to alleviate thermal instability in photosystem II, or preferentially degrade other thylakoid membranes (i.e., PGs, SQs, or MGs) in response to heat stress, leaving the remaining IPL pool relatively enriched in DGs. While several individual MG and SQ molecules did demonstrate linear responses to temperature (Fig. 3), other stressors such as N availability, pH, and light levels may confound the effects of temperature in the overall abundance of these lipid classes. Additional detailed analyses of core fatty acids associated with changing temperatures may elucidate the sources of acyl units for TAG production under heat stress.

**4.2.5 Light Availability**

Bach et al. (2020) considered the overall productivity in these mesocosm experiments to be co-limited by N and light availability, which have been identified as key limiting factors in eastern boundary upwelling systems (Messié and Chavez, 2015). In our experiment, initial high biomass productivity led to self-shading effects that significantly reduced the PAR (Bach et al., 2020). While the maximum photon flux densities measured at noon over the course of the experiment were ~500 – 600 µmol m$^{-2}$s$^{-1}$, only ~5-22% and < 4% were measured in surface (~ 2m) and subsurface (17m) waters, respectively, from all four mesocosms. While both depth-integrated samples in our experiment are likely from light-limited planktonic communities, most PAR values (> 15% i.e., 75-132 µmol m$^{-2}$s$^{-1}$) in surface waters still demonstrate sufficient levels that can dramatically



'16    affect lipid production and accumulation (Gonçalves et al., 2013; Yeesang and Cheirsilp, 2011; Jiang et al., 2011), especially

'17    under the combined effects of N-deprivation (Yeesang and Cheirsilp, 2011; Jiang et al., 2011).

'18

'19    Light levels are identified by the Random Forest analyses to be a significant variable in the prediction of MGs, SQs, PGs, and

'20    PCs (Figs. 4, 6, and 7). While few studies have explored the direct effects of irradiance levels on IPL headgroup distributions,

'21    experimental cultures of *Tichocarpus crinitus* found increased IPL production of SQs, PGs, and PCs (and decreased production

'22    of MGs) amongst shade-grown algae (Khotimchenko and Yakovleva, 2005), with greater TAG production under high light

'23    intensity (Khotimchenko and Yakovleva, 2004; Hu et al., 2008). TAG production under higher light intensity may serve as a

'24    mechanism to prevent photochemical damage of algal cells (Roessler, 1990). Over-production of reactive oxygen species

'25    under stress may also cause photoinhibition and damage to membrane lipids, amongst other macromolecules (Hu et al., 2008).

'26    The low yield of structural lipids relative to chl-a concentrations in surface waters (Fig. S5) could potentially signal TAG

'27    accumulation as active algal growth generally favors the production of structural components (Mock and Gradinger, 2000).

'28

'29    The PC lipid class, which is primarily found in extrachloroplast membranes (Mimouni et al., 2018), consistently exhibits

'30    greater relative abundance in the subsurface compared to surface waters. We suggest that light levels at depth likely play a role

'31    (albeit a secondary one, see Fig. S3B) in non-chloroplast IPL production when algal cells are configured for rapid

'32    growth/reproduction. Conversely, under high light as well as under the combined effects of DIC and N limitation at the surface,

'33    algal cells are more prone to survival responses such as TAG production at the expense of IPLs.

'34

'35    Notably in *Tichocarpus crinitus* cultures, MGs, one of the most abundant thylakoid lipids, shows increased abundances under

'36    high light (Khotimchenko and Yakovleva, 2005). Our results show both MGs and SQs in greater relative and absolute

'37    abundances in the surface samples where light is less limited or potentially inhibitive at times, suggesting a significance in

'38    thylakoid lipid compositions to maintain efficient photosynthetic rates under changing light conditions. MGs have been shown

'39    to carry out important photoprotective and antioxidant mechanisms in diatoms (Wilhelm et al., 2014), potentially explaining

'40    their higher abundance at the surface and the significance of light in the Random Forest models. While the exact function of

'41    SQs in thylakoid membranes remains unclear, it has been observed to act as an antagonist to the aggregating action of MGs

'42    (Goss et al., 2009), with a disaggregating effect on light harvesting complexes (Wilhelm et al., 2014). The latter potentially

'43    reflects planktonic responses to the variable PAR (5-22%) in the surface via adjustment of the proportions of SQs and MGs in

'44    the thylakoids.

'45

'46    PG can be found in extrachloroplast membranes but is also an essential component of photosystems I and II, with important

'47    roles in electron transport processes (Sakurai et al., 2003; Wada and Murata, 2009; Kobayashi et al., 2017). Its abundance over

'48    time is also one of the most variable at both sampling depths (Fig. 2D). The effect of light on PGs concentration is inconsistent,

'49    with no clear pattern emerging between relatively high/low PAR; however, several PG moieties show statistically significant



'50    positive linear relationships with light levels. The overall abundance of PGs contributing to the total IPL pool may be affected

'51    by varying production of certain non-chloroplast membranes under lower light conditions, and other thylakoid membranes

'52    under high light conditions, possibly explaining the highly variable overall contribution of PGs to the IPL pool. Additional

'53    analyses of specific fatty acid structures from intact PGs may illuminate sources within the cell, i.e., thylakoid PGs and

'54    extrachloroplastic PGs.

'55

'56    Light intensity has also been shown to have contradictory impacts on fatty acid composition, with some analyses observing

'57    increased unsaturation under high light intensity (Liu et al., 2012; Pal et al., 2011). Others observed that excess light energy

'58    induces synthesis of saturated (SFAs) or monounsaturated (MUFAs) fatty acids in several algal species to prevent

'59    photochemical damage, or the synthesis of polyunsaturated fatty acids (PUFA) for the maintenance of photosynthetic

'60    membranes under low light conditions (Khoeyi et al., 2012, Fabregas et al., 2004, Sukenik et al., 1993; Orcutt and Patterson,

'61    1974). We see little evidence of a direct impact of light intensities on fatty acid composition in the multiple linear regressions,

'62    except for several PG moieties with 2 or more unsaturation showing a positive relationship with light intensity (Fig. 3). This

'63    may be related to differences in optimal irradiance levels between algal groups overlapping to confound any clear patterns.

'64

'65    Given that the normalized PAR in surface waters remains <25%, and that light appears as a variable of secondary importance

'66    in the random forest models, our results suggest that under these experimental conditions, light availability may amplify the

'67    effects of other environmental stressors (i.e., nitrogen availability, oxygen concentration, temperature, and DIC concentration).

'68    We suggest that the combined stressors may result in TAG synthesis at the expense of IPL production, or through direct

'69    degradation of IPLs. The effects of self-shading from high biomass production early in the experiment, when greater nutrient

'70    availability was high, likely contributed to the markedly different IPL distributions observed between surface and subsurface

'71    waters. As a result, the fluctuations in PAR play a role in the observed variations in the relative proportions of prevalent

'72    thylakoid membrane IPLs, including MGs, SQs, and PGs.

'73

'74    **4.3 Implications of IPL Remodeling in Phytoplankton Membranes**

'75    We expect the ongoing changing conditions in the ETSP, such as rising temperatures, ocean acidification, expanding ODZs,

'76    and shifts in upwelling-driven nutrient supply, to instigate multiple physiological responses amongst phytoplankton

'77    communities. IPL remodeling and the reallocation of resources among algae are likely to have cascading effects on the

'78    composition of phytoplanktonic communities, the nutritional value of algal biomass to higher trophic levels, and the cycling

'79    of organic carbon and nitrogen in the upper ocean. Here, we explore the potential consequences of these physiological

'80    responses to environmental conditions and outline future areas of investigation to better constrain the impacts of IPL

'81    remodeling on marine biogeochemistry.

'82



### 4.3.1 Relative adaptability of Phytoplankton Classes to environmental change

IPL classes are either partitioned to the thylakoid membranes (i.e., MG, DG, SQ, and PG) in variable distributions to maintain photosynthetic function, or to extraplastidic membranes under growth/reproductive phases (BL, PC, PE, and PG). As environmental conditions change, the ability of phytoplankton to adjust these IPL compositions may be an important driver controlling the phytoplankton community structure. Within a few days of the ODZ water addition, the mesocosms became oligotrophic, which coincided with a major shift in phytoplankton groups, from predominantly diatoms, Chlorophyceae, and Cryptophyceae to a dominance of 'survivalist' mixotrophic dinoflagellates. An obvious advantage of mixotrophic dinoflagellates to the largely N-limited conditions of this experiment is their ability to scavenge N from the PON pool, and a lower dependence on light compared to the other previously mentioned phytoplankton groups.

To contextualize the impacts of IPL remodeling on the phytoplanktonic community, we performed an additional Random Forest analysis to identify the most important IPLs (both individual moieties and polar head group totals) in the prediction of individual phytoplankton classes. We applied this analysis to identify what IPL remodeling processes may be more readily available to each major phytoplankton class in this experiment (Fig. 8). While all the phytoplankton classes are likely to produce each IPL class to varying extents, the differences in their relative distributions may provide insight into differences in phytoplankton adaptability under changing environmental conditions. Dinoflagellates are heavily associated with high abundances of the glycolipids SQ and DG, with far less apparent dependence on phospholipids and MGs (Fig. 8). The dominance of dinoflagellate biomass is attributed to the severe scarcity of inorganic nitrogen and relatively low light intensity due to self-shading effects (Bach et al., 2020). Dinoflagellates may also take advantage of the recycling of N-bearing PCs and BLs to alleviate nitrogen limitation as well as to produce energy storing TAGs. The high proportion of DGs and SQs and recycling of MGs may also prove advantageous under high pH/low $pCO_2$ conditions in surface waters, alleviating pH stressors, or the energy investment in active $HCO_3^-$ transport for photosynthesis. These IPL remodeling strategies may have played a role in the dominance of dinoflagellates under the post-upwelling experimental conditions.

In the early phases of the experiment and when inorganic N is readily available, diatoms (and to a lesser extent Cryptophyceae and Chlorophyceae) dominate the water column biomass. Each of these phytoplankton classes (as well as *Synechococcus*) indicates a greater dependence of the N-bearing PCs and/or BLs, potentially signaling a reduced capacity to scavenge these extrachloroplastic IPLs within the cell for N, or as substrates for TAG production. Notably, these phytoplankton consistently make up a greater proportion of the total chl-a pool in subsurface waters, suggesting an ability to accommodate the lower $O_2$ concentrations and pH in exchange for greater N availability (see summary in Fig. 8). Some of this adaptability may be related to the recycling of MGs for additional acyl units under hypoxia (most likely amongst Chlorophyceae and *Synechococcus*), and/or the generally higher relative production of structural membrane lipids under low light growth conditions (see impact of light discussion above).





In the Eastern Tropical South Pacific (ETSP), upwelling events exhibit a distinctive trend of increasing frequency and intensity, setting it apart from other eastern boundary systems (Abrahams et al., 2021a; Oyarzún and Brierley, 2019). This phenomenon brings about important changes in environmental conditions, including a decrease in sea surface temperature (~0.37°C, average from 1982 to 2019, Abrahams et al., 2021a) alongside oxygen reduction and reduced pH in the coastal zones of the region (Pitcher et al., 2021). Simultaneously, there is an observed boost in primary productivity (Gutierrez et al., 2011; Tretkoff, 2011). The impact of strong upwelling, followed by rapid thermal stratification when upwelling subsides, could signify abrupt changes that affect the community structure of phytoplankton (Gutierrez et al., 2011). Such environmental changes favor species better adapted to stress conditions, such as mixotrophic dinoflagellates and silicoflagellates, at the expense of diatom populations (Hallegraeff, 2010). A reduction in diatom biomass translates to a decrease in highly nutritious saturated fatty acids for higher trophic levels (Hauss et al., 2012).

Under the ongoing and projected scenarios of climate change, our mesocosm experiment demonstrates how shifts in the phytoplankton community translate into changes in the lipid composition of their cell membranes. This adds weight to the school of thought that lipid remodeling amongst the phytoplankton community could have repercussions at higher trophic levels, as recently discussed by Holm et al. (2022), and on ocean biogeochemistry as we discuss below.

### 4.3.2 Cycling of Carbon, Nitrogen, and Sulfur

Combined, neutral (TAGs) and polar lipids (IPLs) make up on the order of 17.6% to 34.7% of the cell by dry weight under N-replete and N-deprived conditions, respectively (Morales et al., 2021), with other environmental stressors able to compound the accumulation of neutral lipids (Zienkiewicz et al., 2016). This represents a considerable portion of algal cellular carbon, and total water column carbon stocks that supply the transfer of fatty acids to higher trophic levels (Twining et al., 2020). Trophic transfer of fatty acids to grazing zooplankton may be affected by the relative contribution of polar and neutral lipids, as they can have significantly different turnover rates (Burian et al., 2020). This suggests that the relative concentrations of TAGs/IPLs in phytoplanktonic communities may impact the rates of fatty acid remineralization or uptake into higher trophic levels. Additionally, while our analysis did not focus on individual fatty acid structures, other experiments investigating the effects of increasing $pCO_2$ and nitrogen deficiency have demonstrated a reduction in the relative production of essential fatty acids resulting in negative reproductive effects on primary consumers (Meyers et al., 2019). In this context, our IPL results reveal modifications (among polar head groups, carbon numbers, and unsaturations) in response to divergent environmental conditions. However, further investigation of the relative concentrations and turnover rates between TAGs and IPLs, in addition to the production of essential fatty acids are critical next steps in assessing the impacts of IPL remodeling on trophic transfer processes.





TAGs and IPLs are also sources of dissolved organic carbon (DOC) via hydrolysis by bacterial extracellular enzymes
(Myklestad, 2000). Lipids have been observed as the most highly aged component of DOC and are considerably longer lived
than lipids found in suspended POC (Loh et al., 2004). However, much of the DOC pool remains uncharacterized (Nebbioso
and Piccolo, 2012; Hansell and Carlson, 2014) and little is known regarding the specific processes forming refractory DOC
fractions (Hansell, 2013). Furthermore, it is yet unclear as to how DOC cycling may be affected by changing ocean conditions
(Wagner et al., 2020). Yet, preliminary estimates of the total production of relatively refractory DOC (>100-year lifetime)
suggest potential significance to global biogeochemistry (Legendre et al., 2015). Understanding the degradation pathways
involved with abundant lipid pools and how they may contribute to the DOC pool in the surface ocean is of crucial importance
to constraining the fluxes of refractory organic carbon pools.

While IPLs are not a major source of cellular N compared to proteins, the highly labile nature of these molecules after cell
death may be a potential source of rapidly recycled N via bacterial hydrolysis. On average, N-bearing IPLs such as PCs or
BLs, consist of ~2% N by molecular weight. Based on their abundance, they could contribute as much as 10.7 nmol/L of
inorganic N in the highly N depleted surface waters of this experiment. Inorganic N species are exceedingly low, and often
below the limit of detection (< ~0.1 µmol/L, see methods for details), particularly in the surface samples for most of the
experiment after the OMZ water addition. Thus, it is possible that the rapid degradation and recycling of N from IPLs could
prove to be an additional supply of N under extremely limiting conditions. Given the diel variability in TAG production via
eukaryotic phytoplankton and the potential impacts on energy and carbon fluxes (Becker et al., 2020), further analysis is needed
to determine at what temporal scale IPLs may be recycled for TAG synthesis. While Becker et al. (2020) did not find evidence
of diacylglycerol transferase (DGAT) activity related to diel cycles, additional investigations of other transferases or lipases
relevant to the degradation of IPLs remodeling for the purposes of TAG production (e.g., PDAT or PGD) are critical.

IPLs may also be an important intermediate in the surface ocean sulfur cycle. Sulfur metabolites produced by phytoplankton
(Durham et al., 2019) play central roles in microbial food webs by functioning in metabolism, contributing to membrane
structure, supporting osmotic and redox balance, and acting as allelochemicals and signaling molecules (Moran and Durham,
2019). SQs are highly abundant IPLs, particularly in surface waters, and can constitute up to 60 nmolS/L, potentially
contributing to a significant proportion of the dissolved organic S in this region (ranging from ~100-225 nmol/L; Lennartz et
al., 2019). The ability to catabolize SQs is widespread in heterotrophic bacteria and provides a highly ubiquitous source of
sulfur (and carbon) substrates in the synthesis of other essential sulfur metabolites (Speciale et al., 2016). As such, SQs are
considered major contributors to the global biosulfur cycle (Goddard-Borger and Williams, 2017) and their considerable
production in phytoplankton membranes may be augmented by changes in temperature, light availability (i.e., shading effects),
and pH (see discussion above).



## 5 Conclusions

We investigated the potential for phytoplankton to evoke lipid remodeling in response to multiple environmental stressors. Changing oceanographic conditions in the ETSP, as a result of climate change, are expected to result in shifts within the phytoplanktonic communities that drive primary productivity in the surface ocean. We report evidence of multiple environmental variables imparting controls on IPL head groups, which are known to have specific functions in the chloroplast. Temperature significantly impacts various classes of IPL, except for DG and PE, which are differentiated by their structural and energetic roles within the cell, and thus suggest a high sensitivity of the phytoplankton community membranes to thermal stress. These relationships include the production of glycolipids (MG and SQ) in response to thermal stress, the photoprotective and antioxidant mechanisms provided by MGs, and the role of PGs in electron transport processes in photosystems I and II, possibly regulated by light availability. Furthermore, the variations in the abundance of glycolipids such as MG, DG, and SQ between different groups of phytoplankton at the surface compared to the subsurface appear to be regulated by oxygen availability. Given previous work on the role of lipids in cell membranes, our results suggest that IPLs may be used as a substrate for the generation of acyl chains in TAG production in response to environmental stressors such as N limitation, variable pH, hypoxia, and photoinhibition. These cellular reactions broadly contribute to the overall contribution of IPLs to the algal lipid and organic carbon pools as well as the shifting of resources from structural or extrachloroplastic membrane lipids (i.e., PCs, PEs, BLs, and certain PGs) under high growth conditions, to thylakoid/plastid membrane lipids (i.e., MGs, DGs, SQs, and certain PGs) as well as TAGs under limiting or stressing conditions. Finally, we hypothesize that these remodeling processes may involve major shifts in the elemental stoichiometry of the cell, and thus alter the fluxes of C, N, and S to higher trophic levels, signaling potential impacts on the broader cycling of these biorelevant elements under different scenarios of future environmental change.

**Data availability**

The environmental study data are available at: https://doi.org/10.1594/PANGAEA.923395 (Bach et al., 2020). Biomarkers metadata that generates and supports the findings of this study and R code are available in: https://github.com/Guachan/IPLs-KOSMOS/tree/0.1.0 (https://doi.org/10.5281/zenodo.10408453, Cantarero, 2023).

**Author contribution**

JS and SC designed the study. JS funded the study. JS and CV funded sample collection during the mesocosm experiment. UL and LTB designed, funded, organized, and carried out the mesocosm experiment. PA, JS, LTB, and UR carried out sample collection. SIC led laboratory and analytical work with the assistance of ND and under the supervision of JS. SEI carried out data analysis. SIC, JET, BCS, and BR performed statistical analyzes. SIC wrote the manuscript with comments from all co-authors.

**Competing interests**




The authors declare that they have no conflict of interest. At least one of the (co-)authors is a member of the editorial board of Biogeosciences.

**Acknowledgments**

We extend our gratitude to all participants in the KOSMOS Peru 2017 study for their invaluable assistance in mesocosm sampling and maintenance. Our special thanks go to the dedicated staffs from GEOMAR-Kiel and IMARPE for their support throughout the planning, preparation, and execution of this study. We are particularly grateful to A. Ludwig, M. Graco, D. Gutierrez, A. Paul, S. Feiersinger, K. Schulz, J.P. Bednar, P. Fritsche, P. Stange, A. Schukat, and M. Krudewig. We also express our appreciation to the captains and crews of BAP Morales, IMARPE VI, and BIC Humboldt for their support during the deployment and recovery of the mesocosms. We extend special thanks to the Marina de Guerra del Perú, specifically the submarine section of the navy in Callao, the Dirección General de Capitanías y Guardacostas, and the Club Náutico Del Centro Naval. The KOSMOS Peru 2017 took place within the framework of the cooperation agreement between IMARPE and GEOMAR through the German Federal Ministry of Education and Research (BMBF) project ASLAEL 12-016 and the national project Integrated Study of the Upwelling System off Peru developed by the Directorate of Oceanography and Climate Change of IMARPE, PPR 137 CONCYTEC. We acknowledge all members of the Organic Geochemistry Laboratory in addition to K. Rempfert, S. Kopf. T. Marchitto, N. Lovenduski at the University of Colorado Boulder, and M. Long at NCAR, for fruitful discussions.

**Financial support**

This study was funded by the US National Science Foundation CAREER Award 2047057 "MIcrobial Lipidomics in Changing Oceans" (MILCO) to J. Sepúlveda. S. Cantarero and J. Sepúlveda acknowledge additional support from the Department of Geological Sciences and the Center for the Study of Origins at the University of Colorado Boulder. We thank the Millennium Institute of Oceanography (IMO) ICN12_019 through the Agencia Nacional de Investigación y Desarrollo (ANID)—Millennium Science Initiative Program—for providing funding for sample collection. The KOSMOS Peru 2017 was funded by the Collaborative Research Centre SFB 754 Climate-Biogeochemistry Interactions in the Tropical Ocean, funded by the German Research Foundation (DFG) to U. Riebesell. Additional funding was provided by the EU project AQUACOSM through the European Union's Horizon 2020 research and innovation program under grant agreement no. 731065 and through the Leibniz Award 2012, granted to Ulf Riebesell.

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



**Tables**

**Table 1: Concentrations of inorganic nutrients (μmol/L) including dissolved silicate, phosphate, nitrite, ammonium, nitrate, and the associated N:P:Si and pHT in the two deep water batches used in this experiment.**

| Stations | $Si(OH)_4$ | $PO_4^{3-}$ | $NO_2^-$ | $NH_4^+$ | $NO_3^-$ | N:P:Si | pH | Depth (m) |
|----------|-----------|-------------|----------|----------|----------|--------|-----|-----------|
| Station 1 | 17.4 | 2.6 | 0 | 0.3 | 0 | 0.1:1.0:6.7 | 7.47 | 30 |
| Station 3 | 19.6 | 2.5 | 2.9 | 0.3 | 1.1 | 1.7:1.0:7.8 | 7.49 | 70 |

**Table 2: IPL classes included in these analyses and their respective acronyms.**

| IPL Headgroups | Acronym | IPL Classes |
|----------------|---------|-------------|
| Sulfoquinovosyl | SQ | Glycolipid |
| Monogalactosyldiacylglycerol | MG | Glycolipid |
| Digalactosyldiacylglycerol | DG | Glycolipid |
| Phosphatidylglycerol | PG | Phospholipid |
| Phosphatidylethanolamine | PE | Phospholipid |
| Phosphatidylcholine | PC | Phospholipid |
| Diacylglyceryl trimethylhomoserine | DGTS | Betaine Lipid |
| Diacylglyceryl hydroxymethyl-trimethyl-beta-alanine | DGTA | Betaine Lipid |
| Diacylglyceryl carboxyhydroxymethylcholine | DGCC | Betaine Lipid |



**Figures**

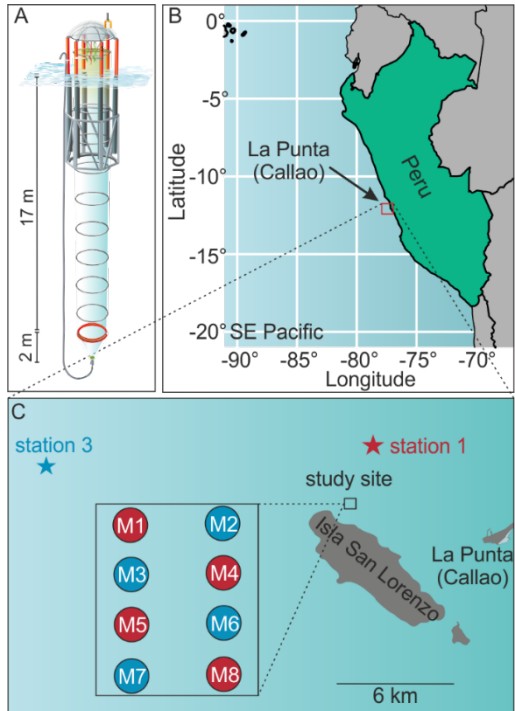

**Figure 1: The KOSMOS study site. (A) Diagram of a mesocosm unit with underwater bag dimensions. (B) Overview map indicating the location of the study region in La Punta, Callao, Perú. (C) Detailed map of the study site with the mesocosm arrangement (M1-8). Red and blue colors indicate the replicates of the two different water treatments. The stars mark the locations of stations 1 and 3, where the two different OMZ water masses were collected. Figure from Bach et al. (2020).**



**Figure 2: Summary of major physicochemical, biological, and lipidomic measurements in surface and subsurface waters over the course of the experiment. A)** Concentration of $O_2$ (μmol/L; left Y axis), and $pH_T$ (total scale; right Y axis). **B)** Concentration of inorganic N (left Y axis), P (left Y axis), and $Si(OH)_4$ (right Y axis) expressed in μmol/L. **C)** Relative abundance (%) of phytoplankton classes (left Y axis) as well as total chl-a concentrations expressed in μg/L (black line; right Y axis). **D)** Relative abundance (%) of major IPL classes based on headgroup contributions to the total IPL pool.



**Figure 3: Summary of multiple linear regression analysis between abundances of major IPL molecules (> 2.5% of total IPL pool) and physicochemical parameters showing only statistically significant (p < 0.05) linear responses. Size of circles indicates the magnitude of the linear regression adjusted $R^2$. Upper and lower panels represent surface and subsurface waters, respectively. Numbers next to circles indicate the total number of carbon atoms and double bonds in core fatty acids chains.**





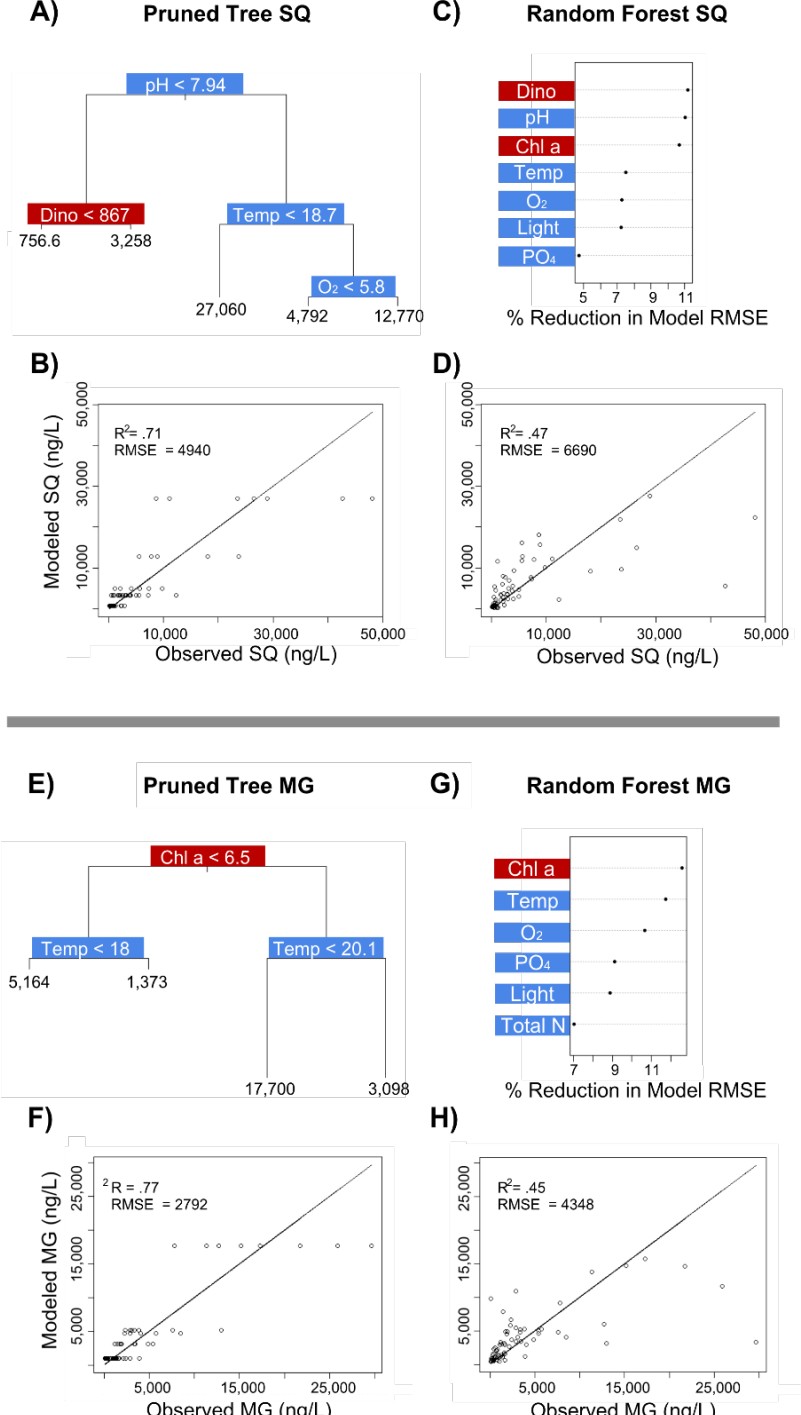

**Figure 4: Classification and Regression Tree (CART) and Random Forest analyses of selected IPL classes (Top: SQ; Bottom: MG). Summary of primary predictors in CART pruned tree (A and E) and CART model performance indicated via adjusted R² and RMSE (B and F). Random Forest variable importance in the prediction of IPL classes as defined by % reduction in RMSE (C and G) and random forest model performance indicated via adjusted R² and RMSE (D and H).**




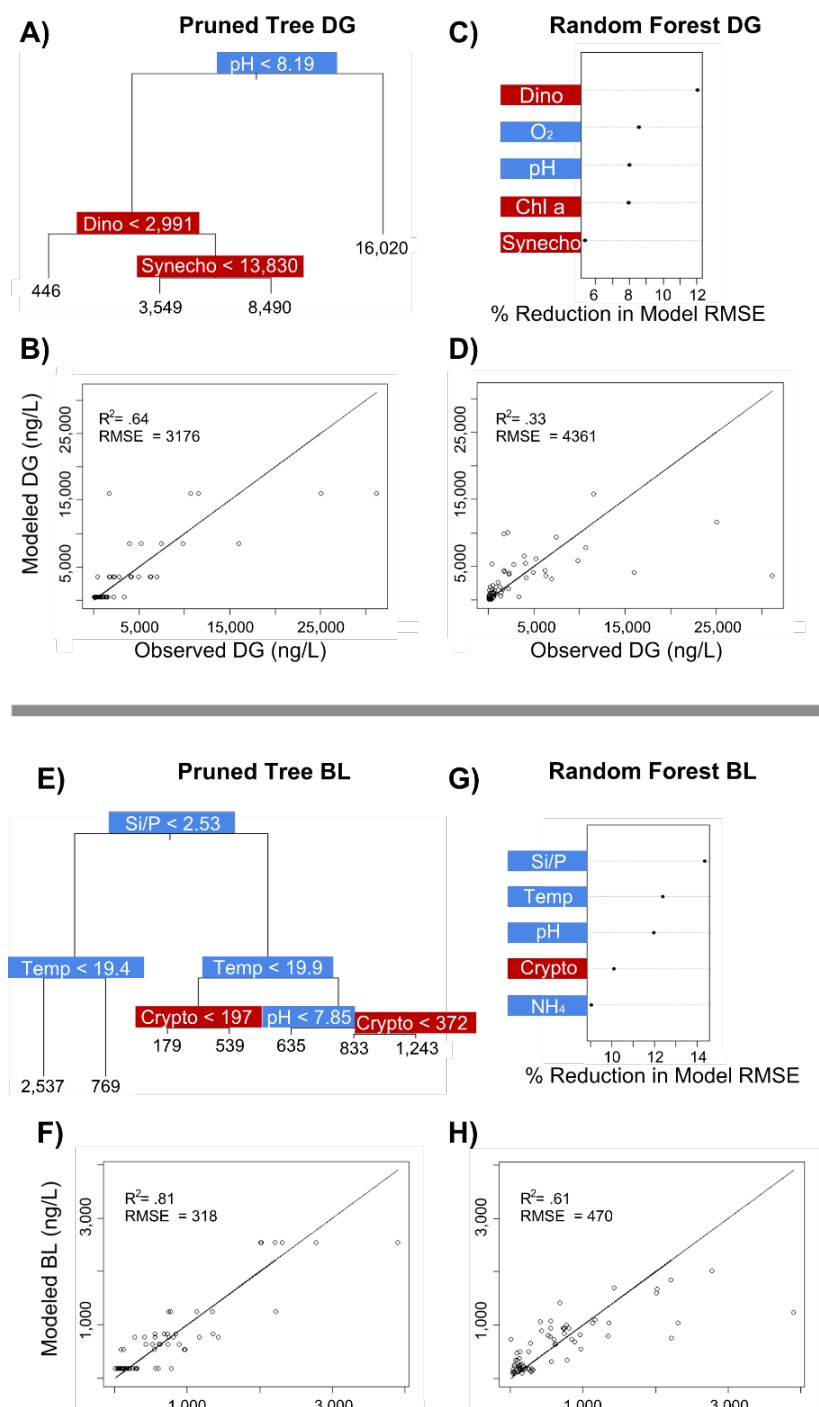

**Figure 5: Classification and Regression Tree (CART) and Random Forest analyses of selected IPL classes (Top: DG; Bottom: BL). Summary of primary predictors in CART pruned tree (A and E) and CART model performance indicated via adjusted $R^2$ and RMSE (B and F). Random Forest variable importance in the prediction of IPL classes as defined by % reduction in RMSE (C and G) and random forest model performance indicated via adjusted $R^2$ and RMSE (D and H).**





**Figure 6: Classification and Regression Tree (CART) and Random Forest analyses of selected IPL classes (Top: PG; Bottom: PE). Summary of primary predictors in CART pruned tree (A and E) and CART model performance indicated via adjusted $R^2$ and RMSE (B and F). Random Forest variable importance in the prediction of IPL classes as defined by % reduction in RMSE (C and G) and random forest model performance indicated via adjusted $R^2$ and RMSE (D and H).**





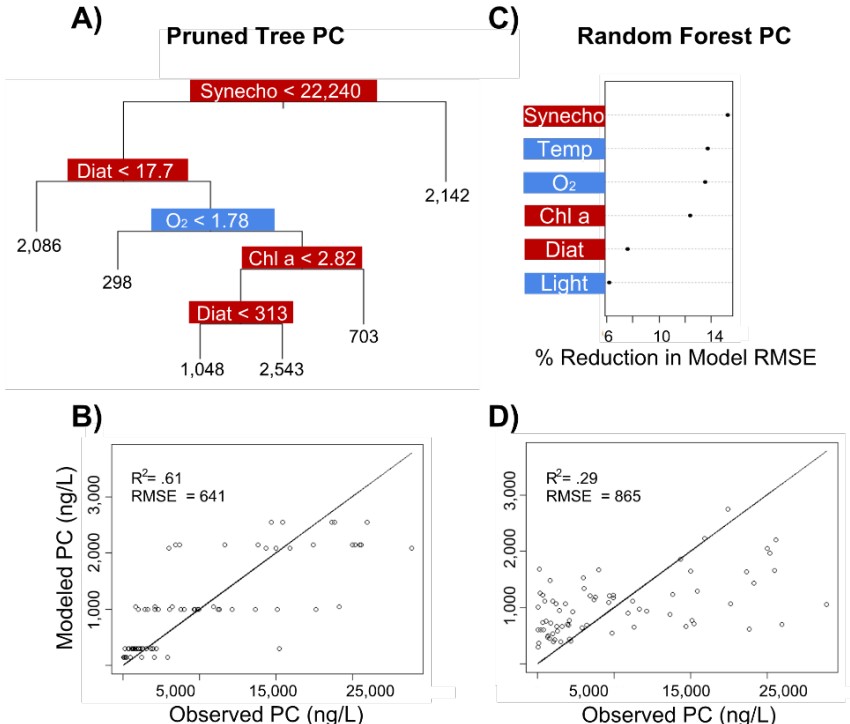

Figure 7: Classification and Regression Tree (CART) and Random Forest analyses of selected IPL class PC). Summary of primary predictors in CART pruned tree (A) and CART model performance indicated via adjusted $R^2$ and RMSE (B). Random Forest variable importance in the prediction of IPL classes as defined by % reduction in RMSE (C) and random forest model performance indicated via adjusted $R^2$ and RMSE (D).





Figure 8: Random Forest based identification of the 15 most important individual IPL molecules or classes in the prediction of phytoplankton classes. Dominant IPL classes and the potential remodeling advantages for each group are summarized. IPL molecules are denoted by their headgroup abbreviation followed by the number of carbon atoms in fatty acid chains, the final number refers to the total number of unsaturations in the fatty acid chains. Root mean square error (RMSE) of random forest model is defined as ng/L of chl-a.