# Peer review of "Lipid remodeling in phytoplankton exposed to multi-environmental"

_EGUsphere, 2023_

## Author Comment (AC1)

Response to Reviewer 1 Comments (RC1):

**We thank the reviewer for the constructive comments**

RC1-1: In this work, Cantarero et al. investigated the lipid remodeling in phytoplankton in response to various environmental variables by mesocosm experiments, including oxygen concentration, temperature, pH, nutrient concentration, chl-a, and light availability. By combining multiple linear regression and random forest model, the main and secondary factors affecting lipid category and lipid distribution were identified. In general, the presented results are interpreted by suitable assessment methods and the novel and important conclusions are supported by the data. In addition, the paper is very well written and fits in quite well with the theme of this special issue, I recommend the manuscript for publication after the following several problems are addressed.

**Response RC1–1: Thank you for your positive review of our work and for the valuable comments to improve this manuscript.**

General comments

RC1-2: What substances can be called intact polar lipids, it is best to give some examples in the introduction section, and contain specific standardized definitions.

**Response RC1–2: We agree and will provide more detail to their specific structural definitions that include the breadth of molecules analyzed in this manuscript.**

**We will add this sentence: "Intact Polar Lipids (IPLs) are a class of membrane lipid characterized by a polar head group typically attached to a glycerol backbone from which aliphatic chains are attached via ester and/or ether bonds (Sturt et al., 2004; Lipp et al., 2008; Schubotz et al., 2009; Van Mooy and Fredricks, 2010). Dominant planktonic lipid classes include phospholipids with a phosphate-bearing polar head group (e.g., phosphatidylcholine PC; phosphatidylethanolamine PE; and Phosphatidylglycerol PG), glycolipids featuring a sugar moiety in the polar head (e.g., monoglycosyldiacylglycerol MG; diglycosyldiacylglycerol DG; sulfoquinovosyldiacylglycerol SQ), and betaine lipids with a quaternary amine positively charged and attached to lipid chains (e.g. diacylglyceryl hydroxymethyl-trimethyl-β-alanine DGTA; diacylglyceryl trimethylhomoserine DGTS; and diacylglycerylcarboxy-N-hydroxymethyl-choline DGCC) (Kato et al., 1997; Rütters et al., 2001; Zink et al., 2003; Suzumura, 2005; Van Mooy et al., 2006)".**

**In addition to their corresponding references:**

**Lipp, J. S., Morono, Y., Inagaki, F., and Hinrichs, K.-U.: Significant contribution of Archaea to extant biomass in marine subsurface sediments, Nature, 454, 991–994, 2008**

**Kato, C., Masui, N., and Horikoshi, K.: Properties of obligately barophilic bacteria isolated from a sample of deep-sea sediment from the Izu-Bonin trench, Oceanogr. Lit. Rev., 1, 53–54, 1997**

Rütters, H., Sass, H., Cypionka, H., and Rullkötter, J.: Monoalkylether phospholipids in the sulfate-reducing bacteria Desulfosarcina variabilis and Desulforhabdus amnigenus, Arch. Microbiol., 176, 435–442, 2001

Suzumura, M.: Phospholipids in marine environments: a review, Talanta, 66, 422–434, 2005

Van Mooy, B. A., Rocap, G., Fredricks, H. F., Evans, C. T., and Devol, A. H.: Sulfolipids dramatically decrease phosphorus demand by picocyanobacteria in oligotrophic marine environments, P. Natl. Acad. Sci. USA, 103, 8607–8612, 2006

Zink K-G, Wilkes H, Disko U, Elvert M, Horsfield B. Intact phospholipids—microbial "life markers" in marine deep subsurface sediments. Org. Geochem; 34: 755-769, 2003

RC1-3: Page 4, Line 36: What is the specific sampling depth of the surface and subsurface layers, and what is the difference between them? There doesn't seem to be an obvious definition.

**Response RC1–3: We agree this information is a bit unclear in the methods sections and we will clarify the usage of terms surface/subsurface and the exact depth intervals they refer to.**

**We will add this sentence: "The samples were segregated into surface and subsurface layers, which were slightly modified over the course of the experiment to accommodate for changes in water stratification and the position of the chemocline (refer to Bach et al., 2020 for further details). The depths were 0–5 and 5–17 m from Day 1 to 2, 0–10 and 10–17 m from Day 3 to 28, and 0–12.5 and 12.5–17 m from Day 29 to 50.**

RC1-4: How efficient is it to use the lipid extraction method described by the author, and has the author conducted relevant validation? Moreover, the analysis conditions of mass spectrometry need to be mentioned appropriately briefly in the methods section, rather than directly citing the literature. What was the detection limit for the various lipids in this study?

**Response RC1–4: The modified Bligh and Dyer extraction method is considered comprehensive in the lipidomic community as it contains 3 different extraction buffers to facilitate dissolution across a large range of analyte polarities and pKa values. We also include a recovery standard ($C_{16}$ PAF $C_{26}H_{54}NO_7P$) to account for potential inefficiencies in the extraction and dilution steps. These details along with the mass spectrometry analysis were originally referenced from a previous publication (Cantarero et al., 2020), but will be added to the methods section in this manuscript for clarity and completeness. We will also report the limit of detection for each lipid class based on individual calibration curves.**

RC1-5: Since I am not an expert in this area, I would like to ask whether the sample number requirements of random forest can be met in this study, and how many sample number were used to conduct it?

**Response RC1–5: We used a total of 72 samples to conduct the random forest analysis. Random forest utilizes a bootstrap aggregation to define and average many permutations of**

**an out of bag score in prediction performance. This provides an effective procedure for high-dimensional data with small sample sizes (Biau and Scornet, 2016) and is popular within a number of related disciplines in the water sciences (Tyralis, 2019) ecological/species distribution models (Luan et al., 2020) and bioinformatics/high throughput genomics (Chen and Ishwaran 2012; Boulesteix et al., 2012). This additional information and the references cited will be included in the methods section.**

RC1-6: Authors would be well advised to standardise the format of journals for references, mostly abbreviations but also full names, e.g. Nature Communications. Please check the format of references in the manuscript.

**Response RC1 – 5: We note that the export of references did not function as intended and will correct them in our revised version of the manuscript.**

RC1-7: Figure 2C: This figure lacks the axis title of the right Y-axis, which is the total chl-a concentrations in µg/L.

**Response RC1 – 7: Thank you for bringing this to our attention. This axis title will be added to the figure.**

Minor comments

RC1-8: Page 3, Line 91: Please check this sentence.

RC1-9: Page 6, Line 78-84: Dichloromethane:Methanol:Phosphate buffer, Dichloromethane / Methanol / Trichloroacetic acid buffer, it is better to unify the two forms. N2 required subscript.

RC1-10: Page 7, Line 11: n = 34in total, lack of space.

RC1-11: Page 22, Line 88: This sentence lacks a full stop. Line 13: 2m and 17m, lack of space.

RC1-12: Figure 8A: Check $R^2$ in the diagram.

**Response RC1 – 8-12: These corrections will be implemented.**

---

## Author Comment (AC2)

Response to Reviewer 2 Comments (RC2):

**We thank the reviewer for their constructive comments**

RC2-1: Review of the manuscript egusphere-2023-3110: „Lipid remodeling in phytoplankton exposed to multi-environmental drivers in a mesocosm experiment" by Cantarero et al.

Using samples from the eastern tropical South Pacific off the coast of Callao, Perú, the authors investigated the lipid remodelling of phytoplankton in the mesocosm in response to various environmental stressors

I have a big complaint about the authors constantly talking about TAGs in the discussion in almost all the subsections, even though they have not analysed them. Especially in the Abstract, the authors go too much into the redistribution of IPL to TAG production in response to various stressors. I think this is unacceptable, especially for the Abstract.

I found the manuscript exhausting to read. There are too many unnecessary discussions.

**Response RC2-1: Thank you for your valuable comments to improve the state of our manuscript. We agree that the discussion could be more concise and we plan to condense the text to mostly focus on the key findings provided by our own data. We also recognize that since we didn't measure TAGs directly we can only hypothesize as to their relation to the IPL distributions observed in this experiment. We will clarify this and limit speculation in the discussion and abstract.**

RC2-2: I suggest that the authors use the common abbreviations for mono- and di-galactosyldiacylglyercols and sulfoquinovosyldiacylglycerol, namely MGDG, DGDG and SQDG.

**Response RC2-2: We recognize that other authors typically use the abbreviations mentioned by reviewer 2 for glycolipids common to phytoplankton. However, all IPLs described in this manuscript contain diacylglycerol structures, so in our opinion, the omission of "DG" in the abbreviations of other phospholipids (eg. PG, PC, and PE) may lead to confusion. We plan, also in response to the comment of reviewer 1 as well, to include more structural information of the IPLs reported in our study in the introduction (see Response RC1–2:). We will continue referring to major phytoplankton IPL classes by their headgroups to provide a simple and consistent naming convention.**

RC2-3: There is unnecessary over citations in the manuscript. The authors have cited 3 or more articles for one statement, one or two would be enough. Consequently, there are too many references (157).

**Response RC2-3: We feel that there is no harm in providing comprehensive citations of relevant and valuable work of our colleagues and predecessors in this field. Furthermore, we are well within the guidelines of the journal regarding citations. We hope that a comprehensive list of references will be helpful for other authors in the future.**

RC2-4: The first digit of the three-digit line numbers is not visible. I hope I have estimated the line number correctly in the comments.

**Response RC2-4: We regret this and appreciate you bringing it to our attention. We suspect that the format change to PDF may have affected the original Word file. We will take this into consideration for the upcoming submission.**

RC2-5: In accordance with the above and the specific comments below, I suggest that the manuscript can be published after Major revision

**Response RC2-5: We appreciate the opportunity to address each of the comments.**

**Abstract**

RC2-6: L 45 – cellular P as well

**Response RC2-6: Thank you, this will be included.**

**1 Introduction**

RC2-7: L 49-50 – In general, three citations are unnecessary. E.g. Ulloa and Pantoja, 2009 and Thamdrup et al., 2012) would be more than enough. This comment applies to all cases where there are 3 or more citations for a statement.

**Response RC2-7: We respectfully disagree with the reviewer. Please refer to the response to comment RC2-3 for further details.**

RC2-8: L 72 – The citation Catalanotti et al. (2013) is inadequate and should be removed.

**Response RC2-8: We agree and will remove it.**

RC2-9: L 79 – All the cell organelles have lipid membrane!

**Response RC2-9: We will address this accordingly in the introduction of IPLs.**

RC2-10: L 80 – I do not find citations Du and Benning 2016; Morales et al., 2021 appropriate.

**Response RC2-10: We agree these references are more pertinent to neutral lipids. Thus, we will remove them and streamline this introductory sentence to focus on relevant literature highlighting specific environmental stressors and their impacts on IPL remodeling.**

RC2-11: L 81 – The citations Urzica et al., 2013 and Gordillo et al., 1998 are missing in the References. I suggest to remove it due to the over citations.

**Response RC2-11: We agree these citations are not essential and will be removed.**

RC2-12: L 82 - Sato et al., 1979 or 1980? Also, the citations Wada and Murata, 1990 and Sinensky, 1974; and Tatsusawa and Takizawa 1996 (first time mentioned) are not appropriate and should be removed.

**Response RC2-12: Thank you, this citation has been corrected to Sato et al., 1980. We agree that Wada and Murata 1990 as well as Tatsusawa and Takizawa 1996 could be removed. We would like to include the earlier work of Sinensky (1974) which laid the foundation for the concept of homeoviscous membrane adaptation.**

RC2-13: L 84-85 - The citation Gombos et al., 2002; Pineau et al., 2004; Simionato et al., 2013; Gašparović et al., are not appropriate and should be removed.

**Response RC2-13: Thank you, we agree that Gombos et al., 2002, Pineau et al., 2004, and Simionato et al., 2013 are misplaced here. We disagree about Gašparović et al. as this study is relevant to the potential impacts of light availability on IPL distribution.**

RC2-14: L 91 - community level and in time series, s, and the associated

**Response RC2-14: This will be corrected.**

**2 Methods**

RC2-15: L 123-124 - It is not clear on which day the ODL water was added. Days 5 and 10?

**Response RC2-15: We will clarify this aspect in the revised version of the manuscript. Days 5 and 10 are when the ODZ waters were collected from stations 1 and 3, respectively. 20 m$^3$ of ODZ water from station 3 was added to mesocosms M2, M3, M6, and M7 on day 11, and 20 m$^3$ of ODZ water from station 1 was added to mesocosms M1, M4, M5, M8 on day 12.**

RC2-16: L 176 – I did not find the protocol for lipid extraction in the paper by Wormer et al. (2013). Therefore, this citation is inadequate.

**Response RC2-16: Thank you for flagging this mistake. We will include the proper citation shown below. Wörmer L., Lipp J. S., Hinrichs K.-U. "Comprehensive analysis of microbial lipids in environmental samples through HPLC-MS protocols" in *Hydrocarbon and lipid microbiology protocols: Petroleum, hydrocarbon and lipid analysis*. eds. McGenity T. J., Timmis K. N., Nogales B. (Berlin, Heidelberg: Springer Protocols Handbooks. Springer), 289–317, 2015**

**3 Results**

RC2-17: L 285, 314, 325, 330, 347, 354, 369, 373, 377, 379 – It would be easier for me to follow the text and Figure 2 if the mesocosms were listed in the order shown in Figure 2. E.g. to say in line 325: ... mesocosms 7, 5 and 8.

**Response RC2-17: We will make this adjustment for improved clarity.**

RC2-18: L 301-302- Fig. 2: The concentrations of the nitrogen species ranged from less than 1 umol/L. The Si concentration in Table 1 is also high (> 17 umol/L), whereas it is not shown in Fig. 2. Is this a question of dilution?

**Response RC2-18: The values reported in Table 1 represent the source ODZ waters before their addition to the mesocosms (~20 m$^3$), whereas Fig. 2 shows the values in the mesocosm throughout the experiment. We will ensure that this is clearly explained in the revised version of the manuscript.**

RC2-19: L 417 – I am not familiar with Card and Random Forest. Perhaps it would be good to include in the Supplements an introduction to understanding Figures 4-7.

**Response RC2-19: We agree that a more extensive explanation of these statistical analyses would be beneficial for readers, which we will provide in the revised version of the manuscript. We plan to add details on the final predictor selection process and the conservative cutoffs that are meant to highlight that only the most significant predictors are included in our final CART and RF figures (Figures 4-7).**

**Regression tree (CART) splitting criteria are determined by evaluating the sum of squared deviations in all possible splits and selecting those that result in the greatest reduction of residual error. In order to prevent overfitting in the CART analysis, a pruning procedure is run to remove nodes that contribute little to the model accuracy based on a cost complexity measure. This procedure allows us to simplify the CART results to focus our interpretations on the most significant predictors of IPL headgroups only. In the Random Forest model, following the averaged cross validated accuracy estimates, we implemented a cutoff of 5% reduction in RMSE to eliminate variables that do not significantly reduce the error of the model prediction. This cutoff allows us to focus our interpretation of only variables that contribute significantly to the out of bag predictor performance.**

RC2-20: My biggest problem is understanding where in Figures 4-7 there is confirmation of what some authors say. I would like an answer to this:

- How can I see that Oxygen concentration was important in predicting MGDSs in Fig. 4E and F?

- How can I see that Oxygen concentration was important in predicting DGDSs in Fig. 5A and B?

- How can I see that temperatue was important in predicting PE predictions in Figs. 6E-H?

- How can I see that Various forms of biologically available nitrogen were important in predicting MGDS in Fig. 4E?

- How can I see that NH4 was important in predicting BLs in Fig. 5E?

- How can I see that NH4 was important in predicting PEs in Fig. 6E-H?

- How can I see that PO4 was important in predicting SQDGs in Fig. 4A?

- How can I see that PO4 was important in predicting MGDGs in Fig. 4E?
- How can I see that light availability was important in predicting SQDGs in Fig. 4A?
- How can I see that light availability was important in predicting MGDGs in Fig. 4E?
- How can I see that light availability was important in predicting PEs in Fig. 6E-H?
- How can I see that light availability was important in predicting PCs in Fig. 7A?

**Response RC2-20: Thank you for making us aware that these important points are not clearly conveyed in the current version of the manuscript. We will expand on the interpretation of these statistical techniques in the methods section as well as in the results and discussion as to aid the reader. Only significant predictors remain in the final CART and RF figures. The pruning and RMSE cutoff procedures described above are intended to remove predictors with little impact on the model performance.**

4 Discussion

RC2-21: L 563 – only Fig. 3

**Response RC2-21: This typo will be corrected.**

RC2-22: L 563-565 – I do not understand this sentence. First the authors state that there is more unsaturated IPL at lower pH values (i.e. in the subsurface), then they state that this is most clearly observed in surface waters???

**Response RC2-22: We agree that this point is unclear and we will clarify the relationship between pH and IPL class proportions and unsaturated moieties. This sentenced will be changed to:**

**The negative correlation between unsaturated IPLs and pH$_T$ (namely amongst glycolipid moieties) is most apparent in surface waters where the variability in pH$_T$ is greatest ($\pm 0.2$).**

RC2-23: L 573-574 – pCO2 is not concentration but the partial pressure of CO2.

**Response RC2-23: This will be corrected.**

RC2-24: L 579 - Hu and Gao et al., 2004 ???

**Response RC2-24: Thank you for bringing this to our attention. We will include the proper reference below: Hu, H. and Gao, K.: Optimization of growth and fatty acid composition of a unicellular marine picoplankton, Nannochloropsis sp., with enriched carbon sources, Biotechnol. Lett., 25(5), 421–425, doi:10.1023/A:1022489108980, 2003.**

RC2-25: L 581-618 – The authors have devoted 4 paragraphs to the discussion of the TAG synthesis. I agree that it's fine to assume that TAG accumulates under unfavourable conditions,

but considering that they did not analyse TAG, I think such a long discussion on TAG is an exaggeration.

**Response RC2-25: We agree with the reviewer and appreciate this comment. We will condense the discussion around TAGs in the revised version of the manuscript and reduce speculation.**

RC2-26: L 608-609 – I suggest that the authors consider why dinoflagellates are more dominant in the surface layer. Dinoflagellates are possibly mixotrophic, some are also heterotrophic, while diatoms are probably limited by Si availability.

**Response RC2-26: Whereas we briefly address this point in section 4.3.1, we plan to expand on it in this particular section of the manuscript as suggested. Namely, we will discuss that mixotrophic dinoflagellates scavenging N from the PON pool likely grants them a considerable advantage in N-depleted surface waters. Their presence in the KOSMOS 2017 experiment, particularly in the final phases, has been reported by Bach et al. (2020) and Min et al. (2023).**

**We will include the recent publication in our references:**

**Min, M.A., Needham, D.M., Sudek, S., Truelove, N.K., Pitz, K.J., Chavez, G.M., Poirier, C., Gardeler, B., von der Esch, E., Ludwig, A., Riebesell, U., Worden, A.Z., Chavez, F.P., 2023. Ecological divergence of a mesocosm in an eastern boundary upwelling system assessed with multi-marker environmental DNA metabarcoding. Biogeosciences 20, 1277–1298. https://doi.org/10.5194/bg-20-1277-2023**

RC2-27: L 631 – How is 125-220 µmol/L just 28% more than ~15-75 µmol/L?

**Response RC2-27: This appears to be a typo that we plan to correct, as it should refer to $O_2$ concentration in the surface vs subsurface.**

RC2-28: L 682-683 – Did the authors analysed the influence of temperature on each individual IPL separately?

**Response RC2-28: The influence of temperature was analyzed through the change in unsaturations for the entire set of IPLs. For example, supplemental figure 2 investigates the relationship between the number of unsaturations (weighted average) in all IPL moieties combined, whereas Figure 3 investigates linear relationships for each individual IPL separately. Neither analysis provided evidence for a consistent correlation between the degree of unsaturation and temperature.**

RC2-29: 4.2.5 Light Availability - Considering that the mesocosms had low light conditions and not high light conditions, I think it is unnecessary to discuss other works about high light conditions.

**Response RC2-29: We agree that this section can be significantly reduced and more focused on the impacts of moderate to low light conditions. We do cite works that suggest the levels observed in this mesocosm experiment are high enough to result in significant production of neutral lipids (TAGs) though, and we plan to reduce this discussion given that we do not provide TAG measurements.**

RC2-30: L 740 - 741 – The authors should read more recent works on the role of SQDG, e.g. DOI: 10.1042/BCJ20170047 ; DOI: 10.1074/jbc.RA118.004304

**Response RC2-30: Thank you for these recommendations. We will review and include them in the revised discussion of SQDG's function in the photosynthetic apparatus.**

**Gabruk, M., Mysliwa-Kurdziel, B., and Kruk, J.. MGDG, PG and SQDG regulate the activity of light-dependent protochlorophyllide oxidoreductase. Biochem. J. 474, 1307–1320. doi: 10.1042/BCJ20170047, 2017.**

**Nakajima, Y., Umena, Y., Nagao, R., Endo, K., Kobayashi, K., Akita, F., Suga, M., Wada, H., Noguchi, T., and Shen, J.-R. J. Biol. Chem., 293, 14786–14797, 2018.**

RC2-31: L 766-767 – I assume it should be written: ... LOW light availability may amplify the effects of other environmental stressors…

**Response RC2-31: This will be corrected.**

RC2-32: L 783 - 4.3.1 Relative adaptability of Phytoplankton Classes to environmental change: Most of the text in this subsection, including Fig. 8, should be moved to the Results section.

**Response RC2-32: We use the RF model in this subsection to highlight the most predictive lipid moieties and lipid classes for each phytoplankton class, and to suggest what their most likely dominant biological sources are. This subsection and Figure 8 are meant to provide an overview of the IPL remodeling strategies potentially available to the major phytoplankton classes of this experiment and to hypothesize as to how these remodeling strategies may play a role in their distribution in this environment. We prefer to leave this section in the discussion as it provides an overview relating the interplay of IPL distributions, phytoplankton distributions, and changing environmental conditions.**

**Figures**

RC2-33: Fig. 2c – The legend of this figure lacks an explanation for the black line (chl a).

**Response RC2-33: This will be corrected.**

RC2-34: I suggest to change Chl-a y-axe from 0-10.

**Response RC2-34: Unfortunately, there are several high Chl-a observations that would be removed with this y axis limit. We prefer to keep the axis limits as they are to prevent the omission of data.**

RC2-35: Fig. 2d – it would be better to organise the IPLs according to their cellular origin. I would suggest the following order: MGDG, DGDG, SQDG, PG, PC, PE, BL and others.

**Response RC2-35: We will modify this accordingly to improve clarity.**

**References**

RC2-36: In some references the name of the journal is written with the full name, in others with an abbreviation.

**Response RC2-36: We will correct these inconsistencies in the citations export.**

RC2-37: The reference Jiang and Jónasdóttir should be separated from the reference Hutchins and Jiang

RC2-38: Reference Guckert, J. B., and Cooksey, K.E.: to correct: …high Ph-induced... DOI: 10.1111/j.0022-3646.1990.00072.x

RC2-39: Reference Schubotz, F., Xie, S., Lipp, J. S., Hinrichs, K. and Wakeham, S. G.: is poorly written.

RC2-40: Also, The reference "Schubotz, F., Xie, S., Lipp, J. S., Hinrichs, K. and Wakeham, S. G" should be separated from the reference Shulz and Riebessel

RC2-41: Reference Cantarero should be improved: ... Front. Mar. Sci. 7:540643. doi: 10.3389/fmars.2020.540643, 2020...

**Response RC2-37-41: These will be corrected.**

---

## Author Comment (AC3)

Response to Reviewer 3 Comments (RC3):

**We thank the reviewer for the constructive comments**

Cantarero Biogeosciences Notes

RC3-1: Summary:

The authors use a combination of multiple linear regression, random forest algorithms, and classification trees to interrogate how physicochemical environmental factors and phytoplankton community structure drive the bulk intact polar lipid pool.

The authors have combined an impressive array of approaches to tease apart a complex system, and the application of these methods to environmental lipidomics is relatively novel (though this paper makes its difficulties clear). Models like these are an important aspect of how the field is currently thinking about interacting environmental drivers, and, with some work, this paper could shed light on how these variables influence the community lipidome and how the lipidome may influence community structure. The paper was generally well written, though the discussion was overly-long and I often felt like it lacked direction, so I would recommend reducing the amount of speculation in the discussion, especially with regards to unsubstantiable TAG production.

**Response RC3-1: We thank the reviewer for their positive and constructive comments that will allow us to revise and improve our manuscript.**

RC3-2: These are the areas where I think the paper could be significantly improved:

In the MLR section, the authors performed 1650 statistical tests (10 parameters x 165 IPLs), the results of which drove a significant portion of the discussion, but they did not mention controlling for false discovery rate. Additionally, the authors mentioned many IPL classes showed both positive and negative relationships (e.g. SQDG), confounding deep interpretation and leading to an extremely long discussion with many "possible explanations." Controlling for FDR would likely reduce the number of significant relationships, and may help the authors focus their efforts on more significant and powerful correlations and therefore drivers of IPL abundance.

Due to this lack of clear direction, paragraph and section structure in the discussion was not always clear—by that I mean, frequently, the authors suggest a potential cause of various differential IPL abundances, often with little or no evidence in the data and with a citation of a singly study, and then the idea is dropped after a couple of sentences without a clear tie to an overall narrative. Potential TAG production was often invoked as an explanation, despite no TAG data in the manuscript, and I hesitate to endorse such an overarching conclusion with no evidence to support it.

**Response RC3-2: Thank you for this thoughtful and thorough suggestion. We agree that controlling for FDR simplifies the interpretation of these MLRs and we will include this reanalysis in the edited manuscript. As suggested, it does allow us to reduce the discussion in several instances where both positive and negative correlations previously complicated our**

**interpretations. This provides us an opportunity to streamline the discussion and focus on the most significant linear correlations between IPLs and environmental conditions.**

**We agree that without having measured TAG concentrations directly, the explanation for IPL distributions in several instances is limited and not conclusive. We will reduce the speculation of TAG production in the discussion and focus more so on the observations made directly in this analysis.**

RC3-3: Additionally, there was little to no mention of bacteria, or of IPL abundance as a proportion of particulate organic matter—I am not an expert in the Peruvian Upwelling Zone, so it may be that they are an insignificant proportion of the biomass in this location. However, many of the IPLs in this study are prevalent in bacteria (e.g. PE, PG, DGTS, MGDG), in some cases moreso than in phytoplankton. (Twice, the authors suggest a higher IPL/chl a ratio in the subsurface is indicative of greater IPL contribution to phytoplankton biomass—however, this could be explained by either non-phytoplankton, i.e. bacterial, or non-living/detrital IPL pools.) This inquiry may help the authors ground their interpretations of the community response to environmental stressors. Without this understanding of how phytoplankton IPLs fit into the total organic carbon pool, many of their interpretations are incomplete.

**Response RC3-3: Our previous work in the Humboldt Current System shows that the ratio of total IPLs to POC is highest at the peak of chlorophyll, as is the relative abundance of particle attached IPLs vs free living sources (Cantarero et al., 2020). This suggests that the majority of the biomass (and IPL content) measured at these high chlorophyll depths is likely to be derived from phytoplankton. We note that our mesocosm study compresses the depth of the chlorophyll maximum and oxycline into a 20m deep system, likely resulting in a greater contribution of phytoplankton lipids in ODZ waters than would be expected stretched over a deeper oxycline with reduced light availability. Bach et al. (2020) demonstrated that whereas light intensity decreases quickly with depth in these mesocosms, the chlorophyll a maximum remained in the upper 5m in the week after the ODZ water addition, but shifted to the intermediate depth range of between 5 and 15m thereafter and until day ~40.**

**Despite this, we agree that there is no perfect separation between bacterial and eukaryotic IPLs and that we can not rule out some contribution from the former to our data, which we consider to be predominantly but not exclusively phytoplanktonic in origin. As we highlight in section 2.5, our approach to minimize the contribution of bacterial IPLs includes the removal of molecules with odd and/or short chain fatty acids as well as certain headgroups exclusively found in bacteria). However, we recognize that there is certainly still a bacterial component in these IPL distributions and we will qualify that potential in the edited manuscript when suggesting the evidence for a greater proportion to phytoplankton biomass. This will include a detailed review of bacterial and phytoplankton distributions based on the reports from other publications in this special issue (Bach et al., 2020; Min et al., 2023).**

**We do value this comment as it also highlights the need for environmental lipidomics in order to parse the many biological sources and environmental forcings on the lipid pool. Indeed,**

we have a follow-up manuscript in prep focusing on IPLs thought to be predominantly found in bacterial biomass (following the reverse filtering procedures stated above). We hope to contribute to this growing field in a subsequent publication focused on the biological sources and environmental drivers among bacterial and archaeal plankton.

RC3-4: Finally, throughout the Discussion, the authors suggest potential triggers that induce microbes to alter their IPL pool (e.g. light, pH, temperature, etc). However, the authors then jump to invoking them as the actual cause in the Conclusion, which I do not think is a valid conclusion from correlation analyses—i.e. correlation does not imply causation.

**Response RC3-4: We agree that this is an overstep in the language used to summarize the correlations observed in these analyses and will clarify the degree to which we may assert these claims.**

Specific comments

RC3-5: I hesitate to question data, but in Fig. 2D, Subsurface, on Days 12, 16, and 18, PG comprises almost 75% of all IPL's. I know of no phytoplankton or marine bacteria where PG makes up anywhere near that percentage—I typically think of them on the order of 5-10%, maybe 20%, in phytoplankton (e.g. Cañavate et al. New Phytologist, 2017; Popendorf et al. Org. Geochem. 2011). Anything higher than that would, to me, seem more indicative of marine bacteria, which, as I said, were not mentioned in the manuscript. I would recommend the authors check their quantification calculations.

**Response RC3-5: We agree that these are high relative abundances for PG (up to 65% of the total IPL pool in some extremes) for predominantly phytoplanktonic biomass. We do analyze an exhaustive array of calibration standards to account for ionization differences between every IPL head group quantified in this analysis. In addition, we have several deuterated standards to account for potential matrix effects (including a deuterated PG standard). These method details are referenced in Cantarero et al., 2020 but will also be included in the edited manuscript per the request of Reviewer 1).**

**We recognize as stated in RC3-2 that there is likely some component of bacterial biomass in the form of IPLs that are common to both domains. While many of the more abundant individual moieties in this experiment (e.g. PG 32:0, PG 30:1, PG 34:2, PG 36:1) have shown strong correlations with high chlorophyll concentration in the ETSP (Cantarero et al., 2020), we do agree it is important to clarify that some of these trends could be in part driven by enhanced bacterial contributions to the IPL pool.**

RC3-6: There are many citations in the manuscript where sources are not listed in the bibliography – e.g. Brandsma et al. 2011 (L 670), Abida et al. 2015, Urzica et al. 2013, Gordillo et al. 1998 (all L 81) – there may be more, but these are the ones I happened to check because I was interested.

**Response RC3-6: These will be corrected.**

RC3-7: L 861 – Where does the 2% N come from? (Citation?)

**Response RC3-7: This was an estimation based on the stoichiometry of a typical betaine lipid with a molar mass of ~700 g/mol. We will clarify that we make this estimation in the text.**

---

## Author Response (AR1)

**Response to Reviewer 1 Comments (RC1):**

RC1-1: In this work, Cantarero et al. investigated the lipid remodeling in phytoplankton in response to various environmental variables by mesocosm experiments, including oxygen concentration, temperature, pH, nutrient concentration, chl-a, and light availability. By combining multiple linear regression and random forest model, the main and secondary factors affecting lipid category and lipid distribution were identified. In general, the presented results are interpreted by suitable assessment methods and the novel and important conclusions are supported by the data. In addition, the paper is very well written and fits in quite well with the theme of this special issue, I recommend the manuscript for publication after the following several problems are addressed.

**Response RC1–1: Thank you for your positive review of our work and for the valuable comments to improve this manuscript.**

General comments

RC1-2: What substances can be called intact polar lipids, it is best to give some examples in the introduction section, and contain specific standardized definitions.

**Response RC1–2: We agree and provide more detail to their specific structural definitions that include the breadth of molecules analyzed in this manuscript.**

**We have added the following paragraphs to the introduction section (see from line 82 onwards):**

**"Intact Polar Lipids (IPLs) are a class of membrane lipids characterized by a polar head group typically attached to a glycerol backbone from which aliphatic chains are attached via ester and/or ether bonds (Sturt et al., 2004; Lipp et al., 2008; Schubotz et al., 2009; Van Mooy and Fredricks, 2010). Dominant planktonic lipid classes include phospholipids with a phosphate-bearing polar head group (e.g., phosphatidylcholine PC; phosphatidylethanolamine PE; and Phosphatidylglycerol PG), glycolipids featuring a sugar moiety in the polar head (e.g., monoglycosyldiacylglycerol MG; diglycosyldiacylglycerol DG; sulfoquinovosyldiacylglycerol SQ), and betaine lipids with a quaternary amine positively charged and attached to lipid chains (e.g. diacylglyceryl hydroxymethyl-trimethyl-β-alanine DGTA; diacylglyceryl trimethylhomoserine DGTS; and diacylglycerylcarboxy-N-hydroxymethyl-choline DGCC) (Kato et al., 1997; Rütters et al., 2001; Zink et al., 2003; Suzumura, 2005; Van Mooy et al., 2006)".**

**We inclused the following references:**

**Lipp, J. S., Morono, Y., Inagaki, F., and Hinrichs, K.-U.: Significant contribution of Archaea to extant biomass in marine subsurface sediments, Nature, 454, 991–994, 2008**

Kato, C., Masui, N., and Horikoshi, K.: Properties of obligately barophilic bacteria isolated from a sample of deep-sea sediment from the Izu-Bonin trench, Oceanogr. Lit. Rev., 1, 53–54, 1997

Rütters, H., Sass, H., Cypionka, H., and Rullkötter, J.: Monoalkylether phospholipids in the sulfate-reducing bacteria Desulfosarcina variabilis and Desulforhabdus amnigenus, Arch. Microbiol., 176, 435–442, 2001

Suzumura, M.: Phospholipids in marine environments: a review, Talanta, 66, 422–434, 2005

Van Mooy, B. A., Rocap, G., Fredricks, H. F., Evans, C. T., and Devol, A. H.: Sulfolipids dramatically decrease phosphorus demand by picocyanobacteria in oligotrophic marine environments, P. Natl. Acad. Sci. USA, 103, 8607–8612, 2006

Zink K-G, Wilkes H, Disko U, Elvert M, Horsfield B. Intact phospholipids—microbial "life markers" in marine deep subsurface sediments. Org. Geochem; 34: 755-769, 2003

RC1-3: Page 4, Line 36: What is the specific sampling depth of the surface and subsurface layers, and what is the difference between them? There doesn't seem to be an obvious definition.

**Response RC1–3: We agree this information is a bit unclear in the methods sections, so we have clarified the usage of "surface" and "subsurface" terms as well as the exact depth intervals they refer to.**

**We have added the following paragraphs to the methods section (see from line 143 onwards):**

 **"The samples were collected across two integrated intervals of the water column representing surface and subsurface layers, sampling depths were slightly modified over the course of the experiment to accommodate for changes in water stratification and the position of the chemocline (refer to Bach et al., 2020 for further details). The depths applied for surface and subsurface waters were 0–5 and 5–17 m on days 1 and 2, 0–10 and 10–17 m from day 3 to 28, and 0–12.5 and 12.5–17 m from day 29 to 50."**

RC1-4: How efficient is it to use the lipid extraction method described by the author, and has the author conducted relevant validation? Moreover, the analysis conditions of mass spectrometry need to be mentioned appropriately briefly in the methods section, rather than directly citing the literature. What was the detection limit for the various lipids in this study?

**Response RC1–4: We used the most standard and commonly used extraction method based on the modified version of the original Blight and Dyer methods. Whereas we acknowledge that this method is not perfect, it continues to be the most appropriate one for the extraction of IPLs as summarized in the review book chapter titled *"Comprehensive Analysis of Microbial Lipids in Environmental Samples Through HPLC-MS Protocols"* by Wörmer et al. (2015). We have included specific mention of C16 PAF purchased from**

Avanti lipids, used as an internal standard, and the sentence "the limit of detection for all lipid classes tested, based on individual calibration curves, was determined to be 0.01 ng on column, with the exception of DGTS (0.001 ng) and DG, MG, and SQDG classes (0.1 ng)". Please see revised method section beginning with line 220.

The mass spectrometry method used in this study followed the conditions described in our previous work (Cantarero et al., 2020). We have added more details to the methods section beginning at line 204 to improve clarity and completeness with the following sentences and reference:

"Briefly, a flowrate of 0.4 ml/min was applied to an Aquity BEH Amide column (150 mm, 2.1 mm, 1.7 µm) using a gradient program first described by Wörmer et al 2013. All filtered TLEs were suspended in Dichloromethane:Methanol (9:1, v/v), prior to injection (10 µL) on column. HESI conditions were optimized for maximum intensity across all measured IPL classes with the following conditions: auxiliary gas temperature: 425 °C, capillary temperature 265 °C, spray voltage 3.5 kV, sheath gas flow rate: 35 arbitrary units (AU), auxiliary gas flow 13 AU, S-Lens RF level 55 AU. We also report the limit of detection for each lipid class based on individual calibration curves."

Wörmer, L., Lipp, J. S., Schröder, J. M., Hinrichs, K.U.: Application of two new LC–ESI–MS methods for improved detection of intact polar lipids (IPLs) in environmental samples, Org. Geochem., Volume 59, Pages 10-21, https://doi.org/10.1016/j.orggeochem.2013.03.004, 2013.

RC1-5: Since I am not an expert in this area, I would like to ask whether the sample number requirements of random forest can be met in this study, and how many sample number were used to conduct it?

Response RC1–5: We have added the following text to section 2.7 of the methods (see from line 284 onwards) to add more clarity on this aspect:

"Random forests are derived from bootstrapping of observational data, and the generation of many decision trees based on each bootstrap sample. We employed a total of 72 samples for the random forest analysis which was run separately for each major IPL class (n = 7) to identify the most important environmental predictors (n = 12) for each individual class of lipid. The random forest method utilizes bootstrap aggregation to generate and average numerous permutations of an out-of-bag score in predictive performance. This approach offers an effective methodology for analyzing high-dimensional data with limited sample sizes (Biau and Scornet, 2016). This method has been widely adopted across various disciplines within the water sciences (Tyralis, 2019), ecological and species distribution modeling (Luan et al., 2020), as well as in bioinformatics and high-throughput genomics (Chen and Ishwaran, 2012; Boulesteix et al., 2012)."

New cited references:

Biau G, Scornet E. A random forest guided tour. Test ; 25(2): 197–227, 2016.

**Tyralis, H., Papacharalampous, G., & Langousis, A. A brief review of random forests for water scientists and practitioners and their recent history in water resources. Water, 11(5), 910, https://doi.org/10.3390/w11050910, 2019**

**Chen, X., and H. Ishwaran. Random forests for genomic data analysis. Genomics 99: 323–329, 2012.**

**Boulesteix, A.-L., Bender, A., Lorenzo Bermejo, J., & Strobl, C. Random forest Gini importance favours SNPs with large minor allele frequency: Impact, sources and recommendations. Briefings in Bioinformatics, 13, 292–304, 2012.**

RC1-6: Authors would be well advised to standardise the format of journals for references, mostly abbreviations but also full names, e.g. Nature Communications. Please check the format of references in the manuscript.

**Response RC1 – 6: We note that the export of references did not function as intended and will correct them in our revised version of the manuscript.**

RC1-7: Figure 2C: This figure lacks the axis title of the right Y-axis, which is the total chl-a concentrations in µg/L.

**Response RC1 – 7: Thank you for bringing this to our attention. This axis title has been corrected in the figure.**

Minor comments

RC1-8: Page 3, Line 91: Please check this sentence.

RC1-9: Page 6, Line 78-84: Dichloromethane:Methanol:Phosphate buffer, Dichloromethane / Methanol / Trichloroacetic acid buffer, it is better to unify the two forms. N2 required subscript.

RC1-10: Page 7, Line 11: n = 34in total, lack of space.

RC1-11: Page 22, Line 88: This sentence lacks a full stop. Line 13: 2m and 17m, lack of space.

RC1-12: Figure 8A: Check $R^2$ in the diagram.

**Response RC1 – 8-12: We have addressed and corrected each one of these suggestions in the text.**

**Response to Reviewer 2 Comments (RC2):**

RC2-1: Review of the manuscript egusphere-2023-3110: „Lipid remodeling in phytoplankton exposed to multi-environmental drivers in a mesocosm experiment" by Cantarero et al.

Using samples from the eastern tropical South Pacific off the coast of Callao, Perú, the authors investigated the lipid remodelling of phytoplankton in the mesocosm in response to various environmental stressors

I have a big complaint about the authors constantly talking about TAGs in the discussion in almost all the subsections, even though they have not analysed them. Especially in the Abstract, the authors go too much into the redistribution of IPL to TAG production in response to various stressors. I think this is unacceptable, especially for the Abstract.

I found the manuscript exhausting to read. There are too many unnecessary discussions.

**Response RC2-1: Thank you for your valuable comments. We agree that the discussion could be more concise, and we have condensed the text to mostly focus on the key findings provided by our own data. We recognize that since we didn't measure TAGs directly, we can only hypothesize as to their relation to the IPL distributions observed in this experiment. We have clarified this in the revised version of the manuscript, and we have limited speculation in the discussion to a few hypotheses that remain to be tested in subsequent work. We have also excluded text about TAGs from the abstract.**

RC2-2: I suggest that the authors use the common abbreviations for mono- and di-galactosyldiacylglyercols and sulfoquinovosyldiacylglycerol, namely MGDG, DGDG and SQDG.

**Response RC2-2: We recognize that other authors typically use the abbreviations mentioned by reviewer 2 for glycolipids common to phytoplankton. However, all IPLs described in this manuscript contain diacylglycerol structures, so in our opinion, the omission of "DG" in the abbreviations of other phospholipids (e.g. PG, PC, and PE) may lead to confusion. We also included, in response to the comment of reviewer 1, more structural information of the IPLs reported in our study in the introduction (see Response RC1–2:). We have continued to refer to major phytoplankton IPL classes by their headgroups to provide a simple and consistent naming convention.**

RC2-3: There is unnecessary over citations in the manuscript. The authors have cited 3 or more articles for one statement, one or two would be enough. Consequently, there are too many references (157).

**Response RC2-3: We respectfully disagree with the reviewer, as we feel that there is no harm in providing comprehensive citations of relevant and valuable previous work by colleagues who have paved the way for our work. Furthermore, we are well within the guidelines of the journal regarding citations. We hope that a comprehensive list of references will be helpful for other authors in the future. We have, however, removed a few unnecessary references in some sections.**

RC2-4: The first digit of the three-digit line numbers is not visible. I hope I have estimated the line number correctly in the comments.

**Response RC2-4: We regret this and appreciate you bringing it to our attention. We suspect that the format change to PDF may have affected the original Word file. We will take this into consideration for the upcoming submission.**

RC2-5: In accordance with the above and the specific comments below, I suggest that the manuscript can be published after Major revision

**Response RC2-5: We appreciate the opportunity to address each of the comments and suggestions.**

**Abstract**

RC2-6: L 45 – cellular P as well

**Response RC2-6: Thank you, this has been included.**

**1 Introduction**

RC2-7: L 49-50 – In general, three citations are unnecessary. E.g. Ulloa and Pantoja, 2009 and Thamdrup et al., 2012) would be more than enough. This comment applies to all cases where there are 3 or more citations for a statement.

**Response RC2-7: As we indicated in RC2-3 above, we respectfully disagree with the reviewer regarding the number of citations. We have, however, removed any references that were misplaced or that were not necessary as we detail below.**

RC2-8: L 72 – The citation Catalanotti et al. (2013) is inadequate and should be removed.

**Response RC2-8: We agree and have removed it.**

RC2-9: L 79 – All the cell organelles have lipid membrane!

**Response RC2-9: We have generalized the language in the introduction of IPLs to refer not only to the chloroplast but all organelles.**

RC2-10: L 80 – I do not find citations Du and Benning 2016; Morales et al., 2021 appropriate.

**Response RC2-10: We agree these references are more pertinent to neutral lipids. Thus, we have removed them to streamline this introductory sentence and focus on relevant literature highlighting specific environmental stressors and their impacts on IPL remodeling.**

RC2-11: L 81 – The citations Urzica et al., 2013 and Gordillo et al., 1998 are missing in the References. I suggest to remove it due to the over citations.

**Response RC2-11: We agree these citations are not essential and have removed them.**

RC2-12: L 82 - Sato et al., 1979 or 1980? Also, the citations Wada and Murata, 1990 and Sinensky, 1974; and Tatsusawa and Takizawa 1996 (first time mentioned) are not appropriate and should be removed.

**Response RC2-12: Thank you, this citation has been corrected to Sato et al., 1980. We agree that Wada and Murata (1990) as well as the first mentioning of Tatsuzawa and Takizawa (1996) could be removed, and we have done so. We would like to include the earlier work of Sinensky (1974) which laid the foundation for the concept of homeoviscous membrane adaptation.**

RC2-13: L 84-85 - The citation Gombos et al., 2002; Pineau et al., 2004; Simionato et al., 2013; Gašparović et al., are not appropriate and should be removed.

**Response RC2-13: Thank you, we agree that Gombos et al., 2002, Pineau et al., 2004, and Simionato et al., 2013 are misplaced here and have been removed. We disagree about Gašparović et al., as this study is relevant to the potential impacts of light availability on IPL distribution.**

RC2-14: L 91 - community level and in time series, s, and the associated

**Response RC2-14: This has been corrected.**

**2 Methods**

RC2-15: L 123-124 - It is not clear on which day the ODL water was added. Days 5 and 10?

**Response RC2-15: We have clarified this aspect in the revised version of the manuscript with structural edits of section 2.1. Days 5 and 10 are when the ODZ waters were collected from stations 1 and 3, respectively. 20 $m^3$ of ODZ water from station 3 was added to mesocosms M2, M3, M6, and M7 on day 11, and 20 $m^3$ of ODZ water from station 1 was added to mesocosms M1, M4, M5, M8 on day 12.**

RC2-16: L 176 – I did not find the protocol for lipid extraction in the paper by Wormer et al. (2013). Therefore, this citation is inadequate.

**Response RC2-16: Thank you for flagging this mistake. We have included the proper citation shown below. Wörmer L., Lipp J. S., Hinrichs K.-U. "Comprehensive analysis of microbial lipids in environmental samples through HPLC-MS protocols" in *Hydrocarbon and lipid microbiology protocols: Petroleum, hydrocarbon and lipid analysis*. eds. McGenity T. J., Timmis K. N., Nogales B. (Berlin, Heidelberg: Springer Protocols Handbooks.**

**Springer; ), 289–317, 2015. We note that this citation is a review chapter where the original references for improved versions of the Bligh and Dyer (1959) method are cited.**

**3 Results**

RC2-17: L 285, 314, 325, 330, 347, 354, 369, 373, 377, 379 – It would be easier for me to follow the text and Figure 2 if the mesocosms were listed in the order shown in Figure 2. E.g. to say in line 325: ... mesocosms 7, 5 and 8.

**Response RC2-17: We have made the respective adjustment, consistent with the established order of mesocosms.**

RC2-18: L 301-302- Fig. 2: The concentrations of the nitrogen species ranged from less than 1 umol/L. The Si concentration in Table 1 is also high (> 17 umol/L), whereas it is not shown in Fig. 2. Is this a question of dilution?

**Response RC2-18: The values reported in Table 1 represent the chemistry of the source ODZ waters before their addition to the mesocosms (~20 m$^3$), whereas Fig. 2 shows the chemistry of the mesocosms throughout the experiment. We have clarified this in section 2.1 and also mentioned it in the caption of Table 1 for clarity.**

RC2-19: L 417 – I am not familiar with Card and Random Forest. Perhaps it would be good to include in the Supplements an introduction to understanding Figures 4-7.

**Response RC2-19: We agree that a more extensive explanation of these statistical analyses would be beneficial for readers. We have added details on the final predictor selection process and the conservative cutoffs that are meant to highlight that only the most significant predictors are included in our final CART and RF figures (Figures 4-7).**

**We have added the following paragraph to the methods section 2.7:**

**"Regression tree (CART) splitting criteria are determined by evaluating the sum of squared deviations in all possible splits and selecting those that result in the greatest reduction of residual error. To prevent overfitting in the CART analysis, a pruning procedure is run to remove nodes that contribute little to the model accuracy based on a cost complexity measure. This procedure allows us to simplify the CART results and thus focus our interpretations on the most significant predictors of IPL headgroups only. In the Random Forest model, following the averaged cross validated accuracy estimates, we implemented a cutoff of 5% reduction in RMSE to eliminate variables that do not significantly reduce the error of the model prediction. This cutoff allows us to focus our interpretation of only variables that contribute significantly to the out of bag predictor performance."**

**To view our extended response, please refer to the supplementary material: Note S1.**

RC2-20: My biggest problem is understanding where in Figures 4-7 there is confirmation of what some authors say. I would like an answer to this:

- How can I see that Oxygen concentration was important in predicting MGDSs in Fig. 4E and F?

- How can I see that Oxygen concentration was important in predicting DGDSs in Fig. 5A and B?

- How can I see that temperatue was important in predicting PE predictions in Figs. 6E-H?

- How can I see that Various forms of biologically available nitrogen were important in predicting MGDS in Fig. 4E?

- How can I see that NH4 was important in predicting BLs in Fig. 5E?

- How can I see that NH4 was important in predicting PEs in Fig. 6E-H?

- How can I see that PO4 was important in predicting SQDGs in Fig. 4A?

- How can I see that PO4 was important in predicting MGDGs in Fig. 4E?

- How can I see that light availability was important in predicting SQDGs in Fig. 4A?

- How can I see that light availability was important in predicting MGDGs in Fig. 4E?

- How can I see that light availability was important in predicting PEs in Fig. 6E-H?

- How can I see that light availability was important in predicting PCs in Fig. 7A?

**Response RC2-20: We thank the reviewer for making us aware that these important points were not clearly conveyed in our original submission. We have expanded on the interpretation of these statistical techniques in the methods section to aid the reader. Only significant predictors remain in the final CART and RF figures. The pruning and RMSE cutoff procedures described above are intended to remove predictors with little impact on the model performance.**

4 Discussion

RC2-21: L 563 – only Fig. 3

**Response RC2-21: This typo has been corrected.**

RC2-22: L 563-565 – I do not understand this sentence. First the authors state that there is more unsaturated IPL at lower pH values (i.e. in the subsurface), then they state that this is most clearly observed in surface waters???

**Response RC2-22: We agree that this point is unclear, so we have clarified the relationship between pH and IPL class proportions and unsaturated moieties. The revised version of the manuscript now reads (line 608):**

**"The observed increased proportion of unsaturated IPLs at lower $pH_T$ is most apparent in surface waters where the variability in $pH_T$ is greatest (± 0.2)."**

RC2-23: L 573-574 – pCO2 is not concentration but the partial pressure of CO2.

**Response RC2-23: This has been corrected.**

RC2-24: L 579 - Hu and Gao et al., 2004 ???

**Response RC2-24: The reference has been removed to streamline the discussion.**

RC2-25: L 581-618 – The authors have devoted 4 paragraphs to the discussion of the TAG synthesis. I agree that it's fine to assume that TAG accumulates under unfavourable conditions, but considering that they did not analyse TAG, I think such a long discussion on TAG is an exaggeration.

**Response RC2-25: We agree with the reviewer and appreciate their comment. We have significantly condensed the discussion around TAGs in the revised version of the manuscript and reduced speculation.**

RC2-26: L 608-609 – I suggest that the authors consider why dinoflagellates are more dominant in the surface layer. Dinoflagellates are possibly mixotrophic, some are also heterotrophic, while diatoms are probably limited by Si availability.

**Response RC2-26: Thank you for your suggestion; we have expanded this discussion in section 4.2.1 to more clearly tie in the conclusions of Bach et al., 2020 that this system is likely limited by N availability as opposed to Si. Please see lines (556-563):**

**"The inorganic N:P ratio ranged between 0.13 and 4.67, with higher inorganic N in subsurface waters compared to the surface, and with a N:P minimum reached by day 20. Bach et al., (2020) noted a week after the ODZ water addition increases in the particulate organic carbon to biogenic silica ratios that coincided with low inorganic N and high $Si(OH)_4$ concentrations, suggesting a N-limited system. Thus, we consider this mesocosm system to be overall nitrogen limited with varying degrees of severity. This N-imitation is also reflected in the transition from predominantly diatoms, Chlorophyceae, and Cryptophyceae to a dominance of mixotrophic dinoflagellates approximately 4-6 days after the initial ODZ water treatment (Bach et al., 2020). Such shifts are consistent with their ecological advantage under N-limiting conditions to extract nitrogen from the dissolved organic nitrogen (DON) pool (Kudela et al., 2010)."**

**We have included the recent publication in our references:**

RC2-27: L 631 – How is 125-220 µmol/L just 28% more than ~15-75 µmol/L?

**Response RC2-27: We have clarified the language used to refer to the proportion of glycolipids more clearly, being 28% higher in the oxygenated surface in comparison to the oxygen deficient subsurface. (lines 665-667):**

**"Glycolipids (MG, DG, and SQ) make up on average 28% more of the IPL pool in the oxygenated surface waters (~125-220 µmol/L O2) compared to oxygen deficient subsurface waters (~15-75 µmol/L O2)."**

RC2-28: L 682-683 – Did the authors analysed the influence of temperature on each individual IPL separately?

**Response RC2-28: The influence of temperature was analyzed through the change in unsaturations for the entire set of IPLs. For example, supplemental figure 2 investigates the relationship between the number of unsaturations (weighted average) in all IPL moieties combined, whereas the multiple linear regressions shown in Figure 3 investigates linear relationships for each individual IPL separately. Neither analysis provided evidence for a consistent correlation between the degree of unsaturation and temperature.**

RC2-29: 4.2.5 Light Availability - Considering that the mesocosms had low light conditions and not high light conditions, I think it is unnecessary to discuss other works about high light conditions.

**Response RC2-29: We agree that this section can be significantly reduced and more focused on the impacts of moderate to low light conditions. We do cite previous work suggesting that the light levels observed in this mesocosm experiment are occasionally high enough to result in significant production of neutral lipids (TAGs) though. We have also reduced this overall discussion given that we do not provide TAG measurements.**

RC2-30: L 740 - 741 – The authors should read more recent works on the role of SQDG, e.g. DOI: 10.1042/BCJ20170047 ; DOI: 10.1074/jbc.RA118.004304

**Response RC2-30: Thank you for these recommendations. We reviewed and included them in the revised discussion of SQDG's function in the photosynthetic apparatus.**

**These references were added to the new version of the manuscript:**

**Gabruk, M., Mysliwa-Kurdziel, B., and Kruk, J.. MGDG, PG and SQDG regulate the activity of light-dependent protochlorophyllide oxidoreductase. Biochem. J. 474, 1307–1320. doi: 10.1042/BCJ20170047, 2017.**

**Nakajima, Y., Umena, Y., Nagao, R., Endo, K., Kobayashi, K., Akita, F., Suga, M., Wada, H., Noguchi, T., and Shen, J.-R. J. Thylakoid membrane lipid sulfoquinovosyl-diacylglycerol (SQDG) is required for full functioning of photosystem II in Thermosynechococcus elongatus. Biol. Chem., 293, 14786–14797, 2018.**

RC2-31: L 766-767 – I assume it should be written: ... LOW light availability may amplify the effects of other environmental stressors…

**Response RC2-31: This has been corrected to "low or variable light availability may amplify".**

RC2-32: L 783 - 4.3.1 Relative adaptability of Phytoplankton Classes to environmental change: Most of the text in this subsection, including Fig. 8, should be moved to the Results section.

**Response RC2-32: We use the RF model in this subsection to highlight the most predictive lipid moieties and lipid classes for each phytoplankton class, and to suggest what their most likely dominant biological sources are. This subsection and Figure 8 are meant to provide an overview of the IPL remodeling strategies potentially available to the major phytoplankton classes of this experiment and to hypothesize as to how these remodeling strategies may play a role in their distribution in this environment. We prefer to leave this section in the discussion as it provides an overview relating the interplay of IPL distributions, phytoplankton distributions, and changing environmental conditions.**

**Figures**

RC2-33: Fig. 2c – The legend of this figure lacks an explanation for the black line (chl a).

**Response RC2-33: This has been corrected.**

RC2-34: I suggest to change Chl-a y-axe from 0-10.

**Response RC2-34: Unfortunately, there are several high Chl-a observations that would be removed with this y axis limit. We prefer to keep the axis limits as they are to prevent the omission of data.**

RC2-35: Fig. 2d – it would be better to organise the IPLs according to their cellular origin. I would suggest the following order: MGDG, DGDG, SQDG, PG, PC, PE, BL and others.

**Response RC2-35: We have modified the legend order accordingly to improve clarity.**

**References**

RC2-36: In some references the name of the journal is written with the full name, in others with an abbreviation.

**Response RC2-36: We have reviewed and corrected inconsistencies from the citations export.**

RC2-37: The reference Jiang and Jónasdóttir should be separated from the reference Hutchins and Jiang

RC2-38: Reference Guckert, J. B., and Cooksey, K.E.: to correct: …high Ph-induced... DOI: 10.1111/j.0022-3646.1990.00072.x

RC2-39: Reference Schubotz, F., Xie, S., Lipp, J. S., Hinrichs, K. and Wakeham, S. G.: is poorly written.

RC2-40: Also, The reference "Schubotz, F., Xie, S., Lipp, J. S., Hinrichs, K. and Wakeham, S. G" should be separated from the reference Shulz and Riebessel

RC2-41: Reference Cantarero should be improved: ... Front. Mar. Sci. 7:540643. doi: 10.3389/fmars.2020.540643, 2020...

**Response RC2-37-41: These have been corrected.**

Response to Reviewer 3 Comments (RC3):

RC3-1: Summary:

The authors use a combination of multiple linear regression, random forest algorithms, and classification trees to interrogate how physicochemical environmental factors and phytoplankton community structure drive the bulk intact polar lipid pool.

The authors have combined an impressive array of approaches to tease apart a complex system, and the application of these methods to environmental lipidomics is relatively novel (though this paper makes its difficulties clear). Models like these are an important aspect of how the field is currently thinking about interacting environmental drivers, and, with some work, this paper could shed light on how these variables influence the community lipidome and how the lipidome may influence community structure. The paper was generally well written, though the discussion was overly-long and I often felt like it lacked direction, so I would recommend reducing the amount of speculation in the discussion, especially with regards to unsubstantiable TAG production.

**Response RC3-1: We thank the reviewer for their positive and constructive comments that have allowed us to revise and improve our manuscript.**

RC3-2: These are the areas where I think the paper could be significantly improved:

In the MLR section, the authors performed 1650 statistical tests (10 parameters x 165 IPLs), the results of which drove a significant portion of the discussion, but they did not mention controlling for false discovery rate. Additionally, the authors mentioned many IPL classes showed both positive and negative relationships (e.g. SQDG), confounding deep interpretation and leading to an extremely long discussion with many "possible explanations." Controlling for FDR would likely reduce the number of significant relationships, and may help the authors focus

their efforts on more significant and powerful correlations and therefore drivers of IPL abundance.

Due to this lack of clear direction, paragraph and section structure in the discussion was not always clear—by that I mean, frequently, the authors suggest a potential cause of various differential IPL abundances, often with little or no evidence in the data and with a citation of a singly study, and then the idea is dropped after a couple of sentences without a clear tie to an overall narrative. Potential TAG production was often invoked as an explanation, despite no TAG data in the manuscript, and I hesitate to endorse such an overarching conclusion with no evidence to support it.

**Response RC3-2: Thank you for this thoughtful and thorough suggestion. We agree that controlling for FDR has simplified the interpretation of these MLRs and have included this reanalysis in the revised version of the manuscript. As suggested, this change has allowed us to reduce and streamline the discussion in several instances where both positive and negative correlations previously complicated our interpretations, and thus focus on the most significant linear correlations between IPLs and environmental conditions.**

**We also agree that without having measured TAG concentrations directly, the explanation for IPL distributions in several instances is limited and not conclusive. We have reduced the speculation of TAG production in the discussion and focused more on the observations derived from existing data.**

**Please see the relevant revisions in sections 4.2.1 to 4.2.5 as well as the revised MLR figure below:**

[Figure]

RC3-3: Additionally, there was little to no mention of bacteria, or of IPL abundance as a proportion of particulate organic matter—I am not an expert in the Peruvian Upwelling Zone, so it may be that they are an insignificant proportion of the biomass in this location. However, many of the IPLs in this study are prevalent in bacteria (e.g. PE, PG, DGTS, MGDG), in some cases moreso than in phytoplankton. (Twice, the authors suggest a higher IPL/chl a ratio in the subsurface is indicative of greater IPL contribution to phytoplankton biomass—however, this could be explained by either non-phytoplankton, i.e. bacterial, or non-living/detrital IPL pools.) This inquiry may help the authors ground their interpretations of the community response to environmental stressors. Without this understanding of how phytoplankton IPLs fit into the total organic carbon pool, many of their interpretations are incomplete.

**Response RC3-3: Our previous work in the Humboldt Current System showed that the ratio of total IPLs to POC is high at the chlorophyll maximum and that the composition of IPLs found in surface waters are predominantly consistent with phytoplanktonic biomass (Cantarero et al., 2020). We note that our mesocosm study compresses the depth of the chlorophyll maximum and oxycline into a 20m deep system, likely resulting in a greater contribution of phytoplankton lipids in ODZ waters than would be expected stretched over**

a deeper oxycline with reduced light availability. Bach et al. (2020) demonstrated that whereas light intensity decreases quickly with depth in these mesocosms, the chlorophyll a maximum remained in the upper 5m in the week after the ODZ water addition, although it shifted to the intermediate depth range between 5 and 15m thereafter and until day ~40.

We agree that there is no perfect separation between bacterial and eukaryotic IPLs. Whereas we consider that our IPLs screening separates lipids that are predominantly phytoplanktonic in origin, we cannot rule out a contribution from bacteria to our data. As we highlight in section 2.5, we tried to minimize the contribution of bacterial IPLs by removing molecules with odd and/or short chain fatty acids as well as certain headgroups exclusively found in bacteria. However, we recognize that there is certainly still a bacterial component in these IPL distributions, so we have more clearly acknowledged that potential in the revised version of the manuscript when suggesting the evidence for a greater proportion to phytoplankton biomass (please see section 4.1). We also value this comment because it highlights the need for environmental lipidomics to parse the many biological sources and environmental forcings on the lipid pool. Indeed, we have a follow-up manuscript in preparation that focuses on IPLs thought to be predominantly found in bacterial biomass (following the reverse filtering procedures stated above). We hope to contribute to this growing field in a subsequent publication focused on the biological sources and environmental drivers among bacterial and archaeal plankton.

RC3-4: Finally, throughout the Discussion, the authors suggest potential triggers that induce microbes to alter their IPL pool (e.g. light, pH, temperature, etc). However, the authors then jump to invoking them as the actual cause in the Conclusion, which I do not think is a valid conclusion from correlation analyses—i.e. correlation does not imply causation.

**Response RC3-4: We agree that this is an overstep in the language used to summarize the correlations observed in these analyses and have limited the degree of confidence to which we assert these claims.**

Specific comments

RC3-5: I hesitate to question data, but in Fig. 2D, Subsurface, on Days 12, 16, and 18, PG comprises almost 75% of all IPL's. I know of no phytoplankton or marine bacteria where PG makes up anywhere near that percentage—I typically think of them on the order of 5-10%, maybe 20%, in phytoplankton (e.g. Cañavate et al. New Phytologist, 2017; Popendorf et al. Org. Geochem. 2011). Anything higher than that would, to me, seem more indicative of marine bacteria, which, as I said, were not mentioned in the manuscript. I would recommend the authors check their quantification calculations.

**Response RC3-5: We agree that these are high relative abundances for PG (up to 65% of the total IPL pool in some extremes) to be predominantly assigned to phytoplanktonic biomass. We do analyze an exhaustive array of calibration standards to account for ionization differences between every IPL head group quantified in this analysis. In addition, we have several deuterated standards to account for potential matrix effects (including a deuterated PG standard). These method details are referenced in Cantarero et**

**al. (2020), but we also include them in the revised version of the manuscript per the request of Reviewer 1.**

**As stated in RC3-2, we recognize that there is likely some component of bacterial biomass in the form of IPLs that are common to both domains. While many of the more abundant individual moieties in this experiment (e.g. PG 32:0, PG 30:1, PG 34:2, PG 36:1) have shown strong correlations with high chlorophyll concentration in the ETSP (Cantarero et al., 2020), we do agree it is important to clarify that some of these trends could be in part driven by enhanced bacterial contributions to the IPL pool that are not possible to quantify in this study. We aim to address this aspect in a follow-up manuscript focused on the bacterial community.**

RC3-6: There are many citations in the manuscript where sources are not listed in the bibliography – e.g. Brandsma et al. 2011 (L 670), Abida et al. 2015, Urzica et al. 2013, Gordillo et al. 1998 (all L 81) – there may be more, but these are the ones I happened to check because I was interested.

**Response RC3-6: These have been corrected.**

RC3-7: L 861 – Where does the 2% N come from? (Citation?)

**Response RC3-7: This was an estimation based on the stoichiometry of a typical betaine lipid with a molar mass of ~700 g/mol. We have clarified how we make this estimation in the text.**